# Few-Shot Design Optimization by Exploiting Auxiliary Information

**Arjun Mani** [1]   **Carl Vondrick** [1]   **Richard Zemel** [1]

## Abstract

Many real-world design problems involve optimizing an expensive black-box function $f(x)$, for which Bayesian Optimization is a sample-efficient framework. However, while the basic black-box setting returns a scalar reward, real-world experiments often generate a wealth of useful information. We introduce a new setting where an experiment generates high-dimensional auxiliary information $h(x)$ along with $f(x)$; moreover, a history of relevant, previously-solved tasks is available for accelerating optimization. We develop a novel method based on a neural model which predicts $f(x)$ for unseen designs given a few-shot context containing observations of $h(x)$. We evaluate our method on two challenging domains, robotic hardware design and hyperparameter tuning. On both domains, our method achieves improved few-shot prediction and faster design optimization, outperforming several multi-task optimization methods.

## 1. Introduction

Many real-world design problems involve optimizing an expensive black-box function $f(x)$ over a design space, such as robot hardware design (Liao et al., 2019; Kim et al., 2021), drug discovery (Stanton et al., 2022; Gómez-Bombarelli et al., 2018), or hyperparameter tuning of neural networks (Wang et al., 2024). Bayesian Optimization (BayesOpt) has emerged as a powerful framework for this problem. At a high level, BayesOpt builds a surrogate model $M$ of the objective function. At each iteration, the model's predictions are used to select a design $\mathbf{x}$ of interest, subsequently $f(\mathbf{x})$ is queried, and $M$ is updated with this feedback, repeating until termination when the best design $f(\mathbf{x}^*)$ is returned.

While these methods have proven useful for the basic black-box setting, the setting itself is highly restrictive in com-

parison to modern scientific or engineering experimental setups. In many real-world problems, the experimenter has access to fine-grained observations of the system, or the capacity to measure multiple quantities correlated with the ultimate metric of interest. We might posit such a setting as one where the experimenter receives "extra information" beyond a single scalar reward. For example, in robotic hardware design, a real-world trial may generate a time series of sensor observations along with a final performance measure like grasp stability. Such extra information is presumably useful for understanding *how* one design succeeds while another fails, and could be altered to succeed.

Concretely, we posit an optimization setting where evaluating $\mathbf{x}$ returns both a reward $f(\mathbf{x})$ and some auxiliary (potentially high-dimensional) information $h(\mathbf{x})$, which could be useful for optimization. A common example of $h(\mathbf{x})$ is a time-series of observations $O_1, ...O_t$, such as in the robotics example above. We additionally posit that the (automated) experimenter has access to a history of related tasks $T_1, ...T_n$ with a shared form of the auxiliary information $h$, relevant to solving the current design task. This is similar to a human experimenter who develops intuition about what to observe in an experiment through previous experience, where the form of observations $h$ is most often similar across tasks (e.g. dynamics of a physical system). It is also a common feature in many real-world optimization problems.

Prior work has examined a simpler version of this setting by assuming a single-task with composite structure $f(\mathbf{x}) = g(h(\mathbf{x}))$ and observations of $h(\mathbf{x})$ along with $f(\mathbf{x})$ (Astudillo & Frazier, 2019). Our setting does not strictly assume $h(\mathbf{x})$ arrives from a composite structure, which restricts the form of $h(\mathbf{x})$ and excludes potentially rich auxiliary information correlated with $f(\mathbf{x})$. Moreover, we assume that a task history is available with observations of $h(\mathbf{x})$. Crucially and different from prior work, this requires a design method to generalizably *learn* how to utilize $h(\mathbf{x})$ from the task history, such that it can efficiently optimize new tasks that provide such information. Since $h(\mathbf{x})$ could be high-dimensional and heterogeneous, encoding $h(\mathbf{x})$ in a way that offers insight for new tasks shifts black-box optimization partially to a question of representation learning.

We introduce a novel approach for this new setting. Our approach is to learn a neural model to predict the performance

[1]Department of Computer Science, Columbia University, New York, New York, USA. Correspondence to: Arjun Mani <asm2290@columbia.edu>.

*Proceedings of the 43rd International Conference on Machine Learning*, Seoul, South Korea. PMLR 306, 2026. Copyright 2026 by the author(s).

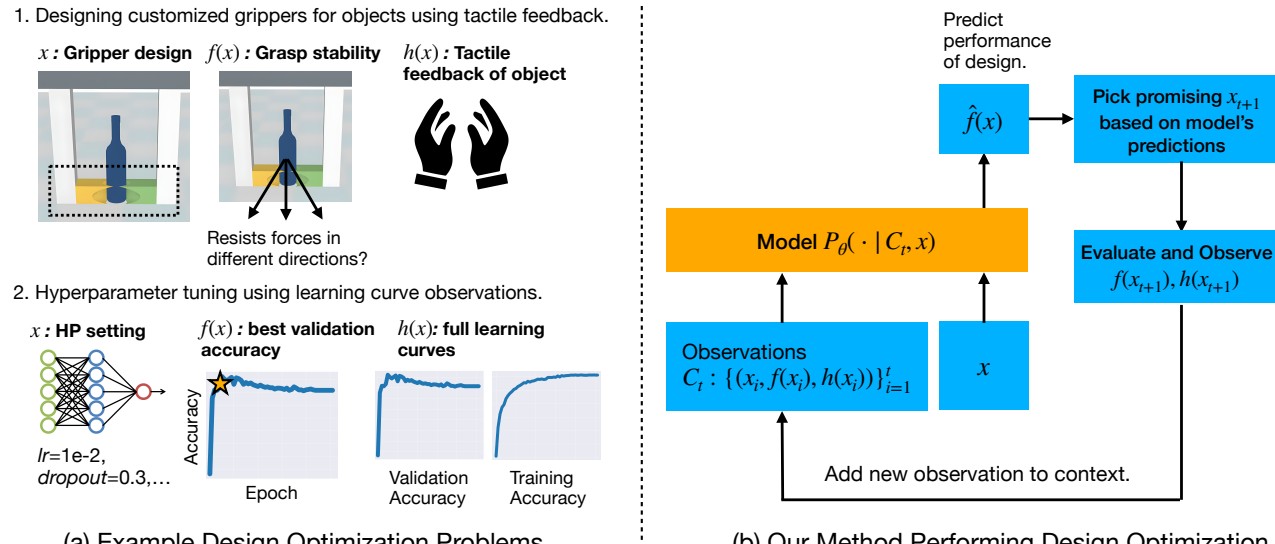

(a) Example Design Optimization Problems.          (b) Our Method Performing Design Optimization.

*Figure 1.* **Few-shot design optimization with our method**. Part (a) shows examples of two design optimization problems in our setting, where for each problem, the design $x$ is indicated, the reward $f(x)$ evaluating design quality is shown, and the auxiliary information $h(x)$ obtained when evaluating a design is indicated. The first problem involves designing a robotic gripper that grasps an object as stably as possible, using tactile feedback during each grasp attempt. In the second problem, evaluating a hyperparameter setting returns per-epoch learning curves as $h(x)$, which can provide useful information beyond the reward (e.g. here, indicating overfitting). Part (b) shows how our method performs design optimization in a loop. Our model $P_\theta$ accepts a few-shot context of observations for a design task, including $h(x)$, and predicts the reward $f(x)$ for an unobserved design. These predictions are used to select a promising new design $x_{t+1}$ for evaluation. The new observation is then added to the model's context, and the next iteration begins. At termination, the design $x^*$ with the highest seen reward is returned. $P_\theta$ is trained on a history of design tasks to acquire this few-shot prediction ability (described in Sec. 4).

of unseen designs given a few-shot context of evaluated designs for a task. Importantly, the auxiliary information is observed in the context, and our method learns how to makes use of $h(\mathbf{x})$ while predicting $f(\mathbf{x})$ for unseen designs. We adopt a transformer-based architecture, which recent work has shown to be effective for few-shot uncertainty-aware prediction (Nguyen & Grover, 2022). The model is trained on a set of tasks with existing evaluations, and applied as a surrogate model for optimization of a new task (Fig. 1).

We evaluate our method on two challenging domains, robot hardware design and neural network hyperparameter tuning. Both domains are shown in Fig. 1. For the robot hardware domain, we create a new design problem for benchmarking methods, based on designing customized robotic grippers for objects using tactile feedback during a grasp attempt. We introduce a large-scale benchmark dataset for this problem. For the second domain, we use per-epoch learning curves as auxiliary information beyond the best validation-set accuracy, and evaluate on the well-known LCBench benchmark (Zimmer et al., 2021). On both domains, our method demonstrates significantly improved few-shot prediction and faster optimization for design tasks unseen during training, compared to several methods for multi-task optimization.

In sum, our work makes three key contributions: *(1)* a new design optimization setting with 'auxiliary information', offering fundamental new challenges, *(2)* a novel method for

this setting, which learns to utilize auxiliary feedback for few-shot prediction, and *(3)* a new gripper design problem and benchmark. Overall, our work takes a step towards more capable systems for AI-driven design, which can operate effectively in realistic scientific or engineering environments.

## 2. Background and Related Work

**Background on Bayesian Optimization.** Bayesian Optimization is a framework for optimization of expensive blackbox functions, which appear often in design. BayesOpt builds a surrogate model $M_\theta$, which makes a probabilistic prediction $M_\theta(\cdot|\mathbf{x})$ for $f(\mathbf{x})$. Given observations of $f$ in a dataset $\mathcal{D}$, the model updates its prior $M_\theta(\cdot|\mathbf{x})$ to a posterior $M_\theta(\cdot|\mathbf{x}, \mathcal{D})$. Popular choices of $M_\theta$ are Gaussian processes (GPs), or neural models like BNNs (Li et al., 2024).

The second component is an acquisition function $\alpha(\mathbf{x})$, which quantifies the value of observing $\mathbf{x}$ based on predictions $M_\theta(\cdot|\mathbf{x}, \mathcal{D})$. A common choice of $\alpha$ is Probability of Improvement: $\alpha_{\text{PI}}(\mathbf{x}) = \mathbb{P}[\hat{f}(\mathbf{x}) > f_{\text{best}}]$, the probability that $f(\mathbf{x})$ exceeds the best-observed value so far. At each iteration, a design $\mathbf{x}_t$ is chosen by solving $\arg\max_{\mathbf{x} \in \mathcal{X}} \alpha(\mathbf{x}, M_\theta)$. Then, $f(\mathbf{x}_t)$ is added to $\mathcal{D}$ and $M_\theta$ is updated. At the end, the best-seen design $\mathbf{x}^*$ is returned.

**Composite Bayesian Optimization**. Extending BayesOpt to include additional information has primarily been studied

under the *composite* setting, which assumes that $f(\mathbf{x}) = g(h(\mathbf{x}))$ and that $h$ is observed along with $f$. Astudillo & Frazier (2019) first studied this setting, and proposed a GP on $h$ with a Monte-Carlo approximation of Expected Improvement as the acquisition function. This work has since been extended by assuming specific forms of $h$ such as intermediate values of nested functions (Astudillo & Frazier, 2021), or to higher-dimensional $h$ via a neural network that maps to lower dimensions (Maus et al., 2024). All these works assume a single-task optimization setting, where any learning is significantly limited by the need for sample-efficient optimization. In our setting, $h$ is assumed to be observed in a history of multiple tasks, and a key challenge is to learn a representation of $h$ that generalizes to a *new* task at test-time. Note also that previous work models $h$ with GPs, which have stricter assumptions and high computational cost, limiting the complexity of representations achievable compared to our neural model-based approach.

**Transfer/multi-task Bayesian Optimization.** Several works have explored the transfer setting in BayesOpt, where a dataset of 'training' tasks with functions $f^{(1)}, ..., f^{(n)}$ are available to accelerate optimization of a new test task with function $f_{\text{test}}$. For each training task, there is a dataset of function evaluations $D^{(i)} = \{(\mathbf{x}_j^{(i)}, f^{(i)}(\mathbf{x}_j^{(i)}))\}_{j=1}^{n_i}$. Approaches for this case divide into putting all data-points into a single surrogate like a multi-task GP (Swersky et al., 2013), learning a mean and kernel prior from training tasks (Wistuba & Grabocka, 2021; Wang et al., 2024), ensembling single-task models (Wistuba et al., 2018), or learning an acquisition function to transfer to new tasks (Volpp et al., 2020). Importantly, the methods in this section do not make use of auxiliary information, and generally capture a logic of 'matching' a test task to the most similar training task.

**Neural Networks for Black-Box Optimization**. Neural networks were first applied to BayesOpt methods for learning the kernel function in GP models (Wilson et al., 2016; Wistuba & Grabocka, 2021; Wang et al., 2024), or less commonly with Bayesian NNs as a surrogate (Snoek et al., 2015; Li et al., 2024). More recently, neural processes (NPs) were proposed as methods that learn the predictive posterior directly from data, by predicting the value of $f(\mathbf{x})$ for a 'target' set from a small 'context' of observations (Garnelo et al., 2018a;b). Recent variants of NPs with transformer-based architectures have shown strong predictive abilities (Nguyen & Grover, 2022; Müller et al., 2022). A final class of end-to-end models directly output the next point to acquire given the observation history (Chen et al., 2022; Maraval et al., 2023), although this requires optimization trajectories at training time or sample-inefficient RL approaches. Our setting challenges methods to incorporate auxiliary information in a generalizable way, presenting a more challenging learning task that requires a novel approach.

## 3. Problem Setting

We formalize our problem setting as follows. We consider a design task $\mathcal{T}$ with a reward function $f(\mathbf{x})$, taking inputs over a design space $\mathcal{X}$. Different from the basic setting, a trial of design $\mathbf{x}$ generates (possibly high-dimensional) auxiliary information $h(\mathbf{x})$ along with $f(\mathbf{x})$. We thus define a trial as a vector-valued function $F(\mathbf{x}) = (f(\mathbf{x}), h(\mathbf{x}))$, and aim to solve the following optimization problem:

$$\mathbf{x}^* \in \arg \max_{\mathbf{x} \in \mathcal{X}} F(\mathbf{x})_0, \qquad (1)$$

where $F(\mathbf{x})_0$ refers to the first element of $F(\mathbf{x})$. Rather than starting to optimize task $\mathcal{T}$ *tabula rasa*, we consider the multi-task/transfer setting where there is a dataset of related 'training' tasks $\mathcal{T}^{(1)}, ..., \mathcal{T}^{(N)}$, associated with functions $F^{(1)}, ..., F^{(N)}$. For each training task $\mathcal{T}^{(i)}$, the dataset contains a set of evaluations $\mathcal{D}^{(i)} = \{(\mathbf{x}_j^{(i)}, F^{(i)}(\mathbf{x}_j^{(i)}))\}_{j=1}^{n^{(i)}}$. These tasks can be used as (pre)training data to learn generalizable features of the task family, applicable to optimizing a new task at test-time, i.e. solving Equation 1.

*Practical Applications.* Many practical problems are encompassed by this setting. In robotics, different robot tasks may require custom hardware or specific tools customized for robot morphology, while a trial of a hardware/tool configuration generates high-dimensional sensor observations (Liao et al., 2019). In drug design, optimizing binding to a target of interest involves *in-vitro* experiments that can measure multiple metrics/attributes of the candidate molecule, while benefiting from experience with related targets (Ramakrishnan et al., 2014). Finding optimal hyperparameter configurations for large neural networks may benefit from full observations of loss curve behaviors, which can suggest phenomena like overfitting (Adriaensen et al., 2023).

*Unique challenges.* Intuitively, while $f(\mathbf{x})$ states the fact of a design's quality or lack thereof, $h(\mathbf{x})$ could provide deeper insight into *how* one design fails while another one succeeds. The presence of a history of related tasks, involving the same form of $h$, challenges design methods to *learn* this deeper insight present in $h$ in a generalizable way, applicable to solving new optimization tasks at test-time. This vital aspect of learning to represent and utilize $h(\mathbf{x})$ is a challenging problem, as $h(\mathbf{x})$ may be complex and high-dimensional, while having a nontrivial relationship with the reward $f(\mathbf{x})$. We note that reasoning about the reward function by way of input similarity (i.e. $||\mathbf{x} - \mathbf{x}'|| \implies |f(\mathbf{x}) - f(\mathbf{x}')|$), which is captured by previous methods, is still essential; however, now equally essential is how to encode $h(\mathbf{x})$. This also indicates the key test of any method that utilizes $h(\mathbf{x})$, that it outperforms any baseline using $f(\mathbf{x})$ alone for optimization.

# 4. Method

We propose a novel method for the setting in Section 3. Our method is based on a neural model which learns to perform few-shot probabilistic prediction, and functions as a surrogate model. Concretely, given a few-shot context of evaluated designs for a task, in which *both* $f(\mathbf{x})$ and the auxiliary information $h(\mathbf{x})$ are observed, the model predicts the reward $f(\mathbf{x})$ for a separate set of unobserved designs. In Sec. 4.1, we detail our foundational approach, agnostic to model architecture. In Sec. 4.2, we describe our transformer-based architecture. In Sec. 4.3, we discuss how our model is used as a surrogate in a BayesOpt procedure.

## 4.1. Foundational Approach

In our setting, we associate a task $\mathcal{T}$ with a vector-valued function $F(\mathbf{x})$, returning both reward $f(\mathbf{x})$ and auxiliary information $h(\mathbf{x})$. Since $f(\mathbf{x})$ and $h(\mathbf{x})$ are correlated, we take the viewpoint of a prior $P(F)$ over the joint function, and consider a task as a sample from this prior $F^{(i)} \sim P(F)$. Conditioning this prior on observations $\mathcal{D} = \{(\mathbf{x}_i, F(\mathbf{x}_i))\}_{i=1}^{n}$ produces a predictive posterior $P(F|\mathcal{D})$. From this viewpoint, we seek a surrogate model $P_\theta$ which can approximate the true prior $P(F)$, or equivalently, approximate the predictive posterior distribution (PPD) such that $P_\theta(F|\mathcal{D}) \approx P(F|\mathcal{D})$.

While $P(F|\mathcal{D})$ could be approximated via a manually specified prior (e.g. a Gaussian Process), this could be conceptually limiting ($h(\mathbf{x})$ may have a complex relationship with both $f(\mathbf{x})$ and $\mathbf{x}$), and computationally intractable ($h(\mathbf{x})$ could be very high-dimensional, e.g. a time-series). Instead, we *learn* a neural network to approximate the PPD of $F(\mathbf{x})$ on a finite dataset, allowing it to learn a useful representation of $h(\mathbf{x})$ in the process. In the basic formulation, the neural network accepts a finite dataset $\mathcal{D}^{(i)}$ as a discrete realization of $F^{(i)}$, assumed to be drawn from $P(F)$. Two disjoint subsets are sampled from this dataset $C, T \subset \mathcal{D}^{(i)}$; the network produces an uncertainty-aware prediction of $F(\mathbf{x})$ for the designs in $T$, given the few-shot observations of $F(\mathbf{x})$ in $C$, this prediction denoted as $P_\theta(\cdot|C, T_\mathbf{x})$. It learns this few-shot prediction ability from a large set of training tasks $\{F^{(i)}\}_{i=1}^{N}$, each implicitly drawn from $P(F)$.

A challenge with this basic formulation is that $h(\mathbf{x})$ can be high-dimensional and potentially heterogeneous. Apart from the computational challenges of predicting $h(\mathbf{x})$, predicting the full content of $h(\mathbf{x})$ may be misaligned with the goal of modeling and optimizing $f(\mathbf{x})$. For example, if $h(\mathbf{x})$ is a time sequence of sensor measurements, only a portion of this sequence may actually be relevant for identifying promising new designs. Accordingly, the model's learning objective should focus on utilizing $h(\mathbf{x})$ for predicting the *performance* of new designs, rather than requiring full reproduction of $h$. Thus, our approach instead models

the posterior over the reward $f(\mathbf{x})$, i.e. $P_\theta(f(T_\mathbf{x})|C, T_\mathbf{x})$, where the context $C$ still contains evaluations of $F(\mathbf{x})$.

We thus set up the following training procedure. The training set consists of tasks $\mathcal{T}^{(1)}, ..., \mathcal{T}^{(N)}$. At each iteration, a task $\mathcal{T}^{(i)}$ is sampled, and two disjoint subsets of $\mathcal{D}^{(i)}$ are sampled to form a context set $C = \{(\mathbf{x}_j, F^{(i)}(\mathbf{x}_j))\}_{j=1}^{N_C}$ and target set $T = \{(\mathbf{x}_k, F^{(i)}(\mathbf{x}_k))\}_{k=1}^{N_T}$. Note that this procedure induces a distribution $Q(C, T)$ over context and target sets. Given observations of $F(\mathbf{x})$ in $C$, the model predicts $f(\mathbf{x})$ for the target designs $T_\mathbf{x}$, and minimizes the following objective:

$$L(\theta) = \mathbb{E}_{C,T \sim Q}[-\log P_\theta(f(T_\mathbf{x})|C, T_\mathbf{x})]. \qquad (2)$$

For simplicity, we assume that the predicted distribution over $f(T_\mathbf{x})$ can be factorized given the context, i.e. $P_\theta(f(T_\mathbf{x})|C, T_\mathbf{x}) = \prod_{\mathbf{x}_k \in T_\mathbf{x}} P_\theta(f(\mathbf{x}_k)|C, \mathbf{x}_k)$. We do not expect that this assumption is too restrictive in practice, but note that it could be relaxed if necessary. For each design $\mathbf{x}_k$ in the target set, the model predicts a univariate normal distribution $\mathcal{N}(\hat{\mu}_k, \hat{\sigma}_k)$ for $f(\mathbf{x}_k)$. In practice, we sample a batch of tasks at each iteration, and average loss across tasks. A pseudocode description is provided in Appendix C.

## 4.2. Model Architecture

We realize $P_\theta$ with a transformer-based architecture. Transformers have several attractive properties for this task: they can produce predictions invariant to the order of context and target sets by choosing a suitable attention structure. Moreover, attention mechanisms can associate a target input $\mathbf{x}$ with relevant context observations of $F$ for predicting $f(\mathbf{x})$.

Our architecture is shown in Figure 2. Each data-point in the context and target sets is considered an individual token for the model. Each token is encoded to a fixed dimension and then passed into the transformer. A small MLP is applied to the last-layer target token representations to predict the reward as a distribution $\mathcal{N}(\hat{\mu}, \hat{\sigma})$. As first proposed in Transformer Neural Processes Nguyen & Grover (2022), we remove positional encodings and use an attention structure where the context points attend to each other, while each target point attends only to the context points.

A crucial difference from prior work is the presence of $h(\mathbf{x})$ in the context, which could be high-dimensional. Therefore, the input encoder requires careful design to ensure that $h(\mathbf{x})$ is represented in an effective way, while not also diluting the influence of $\mathbf{x}$ and $f(\mathbf{x})$. We address this by first adopting separate input encoders $E_\theta^c(\mathbf{x}, f(\mathbf{x}), h(\mathbf{x}))$ and $E_\theta^t(\mathbf{x})$ for the context tokens and target tokens. While the design of the context encoder depends on the form of $h(\mathbf{x})$, we focus on a common case when $h(\mathbf{x})$ is a time-series of observations (Fig. 2b). We encode $h(\mathbf{x})$ itself using a transformer

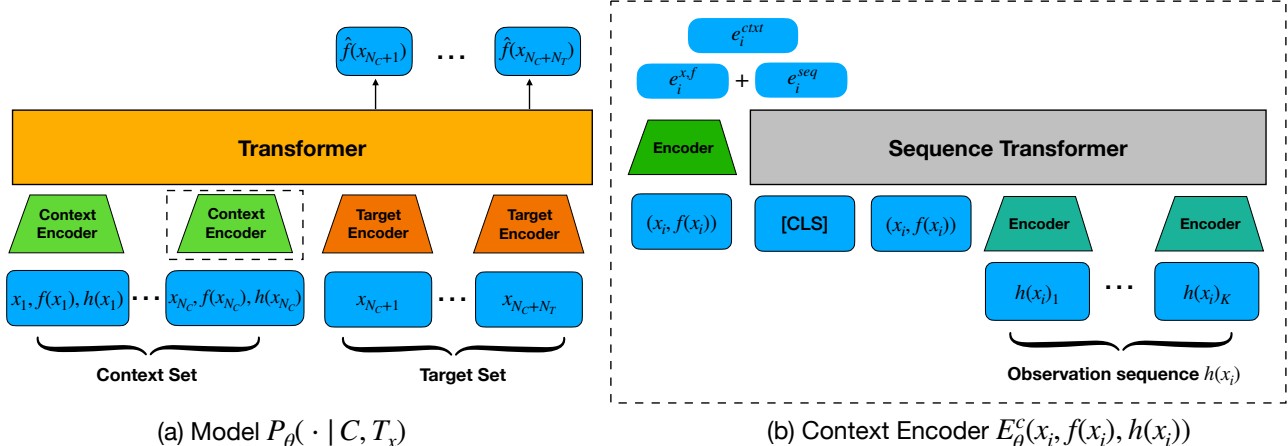

(a) Model $P_\theta(\,\cdot\mid C, T_x)$

(b) Context Encoder $E_\theta^c(x_i, f(x_i), h(x_i))$

*Figure 2.* **Model architecture for our method.** (a) In our model for few-shot probabilistic prediction a small context set of $N_C$ observations is provided as conditioning, in which observations of $f(\mathbf{x})$ and $h(\mathbf{x})$ are available. A target set of size $N_T$ with only inputs $\mathbf{x}$ is also provided, encoded separately. The encoded data-points pass through a transformer where each target point can only attend to the context, and a prediction of $f(\mathbf{x})$ is made for each of the target points. (b) Zooms in to the context encoder that encodes each context point, showing its architecture when $h(\mathbf{x}_i)$ is a temporal sequence of $K$ observations. The temporal sequence, with a token added for $(\mathbf{x}_i, f(\mathbf{x}_i))$, is embedded by a sequence transformer, and added to an embedding of $(\mathbf{x}_i, f(\mathbf{x}_i))$ to obtain the final encoding $e_i^{ctxt}$.

encoder, adding a [CLS] token to obtain a sequence embedding. Note that a special input token encoding $(\mathbf{x}, f(\mathbf{x}))$ is also added, which may influence the valuable parts of the sequence to attend to. Finally, we obtain the embedding for a context observation by adding together the sequence embedding of $h(\mathbf{x})$ and an embedding of $(\mathbf{x}, f(\mathbf{x}))$.

### 4.3. Using our Model for Bayesian Optimization

Once $P_\theta$ is trained, we can use it as a surrogate model in a BayesOpt loop. For a test task $\mathcal{T}$, a small initial set of observations is collected $C_0 = \{(\mathbf{x}_i, F(\mathbf{x}_i))\}_{i=1}^{n_0}$. At each iteration $t$, the next observation is selected by solving the optimization problem $\mathbf{x}_{t+1} = \arg\max_{\mathbf{x}\in\mathcal{X}} \alpha(P_\theta(\cdot|C_t, \mathbf{x}))$. This optimization step can make use of standard, effective acquisition functions like Probability of Improvement (Sec. 2). Then, $F(\mathbf{x}_{t+1})$ is observed and the result is appended to the context to form $C_{t+1}$, repeating until termination. The model itself is not updated during optimization, offering an advantage over methods that require iterative re-training.

In our experiments, $\mathcal{X}$ is discrete and $\alpha$ is optimized over a (large) finite set of designs. However, our method could also be applied to continuous spaces, by differentiating $P_\theta$ w.r.t. the target input and optimizing $\alpha$ via gradient ascent. Pseudocode of this BayesOpt procedure is in Appendix C.

## 5. Experimental Setup

We evaluate our method on two diverse and challenging domains: (1) *robot hardware design* and (2) *neural network hyperparameter tuning*. Both of these domains naturally afford access to auxiliary signals, as time series of sensor observations (the former), or as per-epoch learning curves (the latter). In Sec. 5.1, we discuss the first domain, for which we introduce a new gripper design task and large-scale dataset. We discuss the second domain in Sec. 5.2.

### 5.1. Gripper Design Task

To evaluate methods for our setting, we develop a new black-box design problem, where the aim is to design the shape of robotic grippers using tactile feedback. Many important applications require task-specific grippers customized for grasping specific objects (Burger et al., 2020). Recent works have consequently studied generating customized grippers automatically for different objects (Ha et al., 2021; Xu et al., 2024). We study a variant of this problem where the gripper must be designed via tactile feedback. This has connections to the "blind grasping" problem in robotics, where a robot must grasp an object via tactile feedback alone (an ability that comes naturally for humans) (Dang et al., 2011).

**Task Details**. For this design problem, a task $\mathcal{T}$ involves finding a gripper design $\mathbf{x}$ that can grasp an object $O$ as stably as possible. To evaluate a design, we run a simulation which perturbs the object with disturbance forces (the standard measure for grasp stability (Roa & Suárez, 2015)).

*Simulation Details.* The parallel-jaw gripper consists of a wrist with left and right handles, to which the fingers are attached (see left column, Fig. 3). We parametrize the gripper geometry $\mathbf{x}$ as a cubic Bezier surface, and extrude it to 3D to form the gripper fingers (the left and right fingers are mirrored). We further optimize the initial *height* of the gripper. The total dimension of the design space is 21.

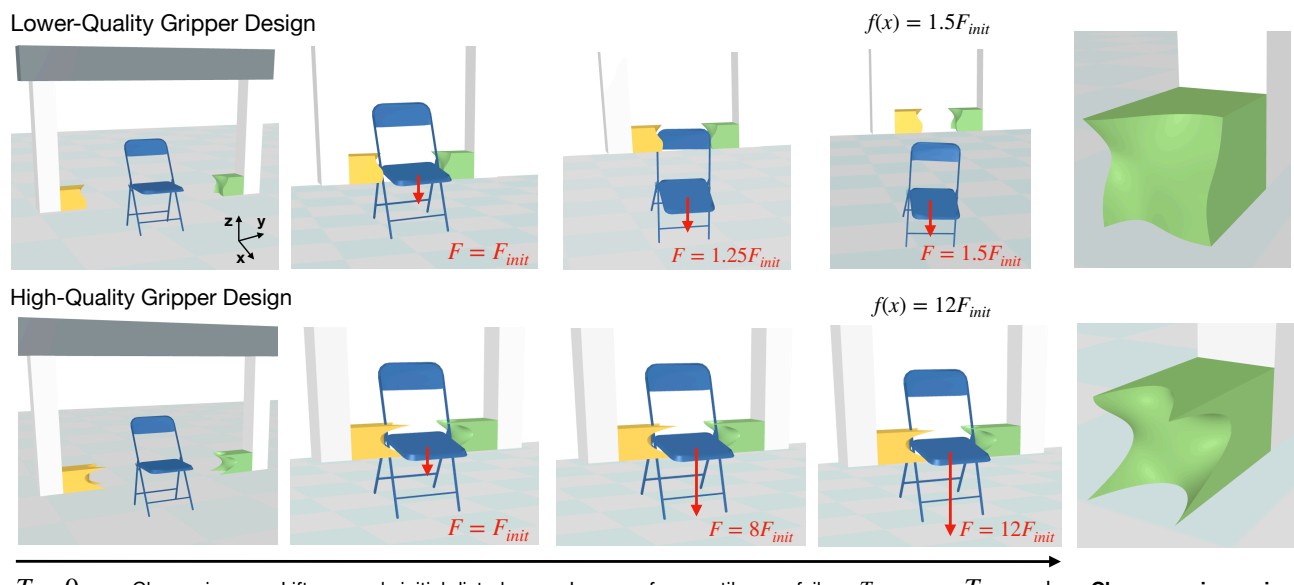

*Figure 3.* **A tale of two grippers: examples of our grasping simulation.** The figure above shows two gripper designs for grasping a chair object, which is a task in the test set. For both, the simulation starts by closing the gripper jaws and lifting up the object to a fixed height. Then, an initial disturbance force $F_{init}$ is applied (here, in the $-z$ direction). As long as the object remains in the grasp, the force is incremented by 0.1N, repeating until some maximum time $T_{max}$. The reward $f(\mathbf{x})$ is the maximum disturbance force applied before the grasp fails, measuring its stability. The top row shows a gripper with a relatively flat surface, which can only resist a mild disturbance before the object falls. Bottom shows a high-quality gripper discovered by our method during optimization. This gripper wraps around both chair legs right under the seat, while also clasping the front leg above the seat, thus resisting any downward or horizontal disturbances. See the close-up view for further clarity on this strategy. It maintains the grasp until the end, achieving greater reward than the first gripper.

During the simulation, the gripper fingers close in and lift up the object to a fixed height (or fail to lift). Then, disturbance forces are applied to the object in a fixed direction. We run three simulations with different force directions to test grasp robustness. Fig. 3 shows the downward direction. For each simulation, we start with an initial force and increment by 0.1N while the object remains in the grasp. The reward $f(\mathbf{x})$ is the maximum force applied before the object falls or the simulation ends, with this averaged over the three simulations. The maximum possible reward is $\sim 6.0$N. We implement the simulation in MuJoCo (Todorov et al., 2012).

*Tactile feedback.* The tactile feedback consists of two 16×16 tactile maps for the inward faces of the left and right grippers, $\text{Ti}_L$ and $\text{Ti}_R$. We also record scalar contact readings for the other faces of each gripper. These tactile readings at each time-step compose the auxiliary information $h(\mathbf{x})$. Further details are in Appendix G.

**Dataset Generation and Statistics**. We sample about 1000 objects from ShapeNet (Chang et al., 2015), a rich collection of 3D models for many common objects. For each object, 400 (unique) random gripper designs are generated, and each gripper design is evaluated at height increments of 0.5 cm until the height of the object is exceeded. These evaluations compose the dataset $\mathcal{D}^{(i)}$ for each of the tasks.

The final dataset consists of 4.28 million designs evalu-

ated across 997 objects. Notably, this benchmark is significant larger than existing multi-task BayesOpt benchmarks (see Appendix F for details; we will make this benchmark publicly-available). Note that most objects in our benchmark have high-performing, stable gripper designs. We divide the dataset into a test set (150 objects), validation set (75 objects), and training set (772 objects). The benchmark is available for download at designopt.cs.columbia.edu.

### 5.2. Hyperparameter Tuning Task

Neural network hyperparameter tuning is a common application for black-box optimization, where BayesOpt methods have proven successful (Turner et al., 2021). These methods generally only utilize a scalar reward (best-achieved validation set accuracy) for optimization. However, neural network training produces full learning curves of key metrics over training time. These learning curves can provide valuable information, for instance whether the model is over-fitting (suggesting stronger regularization, e.g. dropout).

We explore this hypothesis by treating learning curves as auxiliary information. We use the widely-used LCBench benchmark (Zimmer et al., 2021), which consists of 35 HP optimization tasks on classification datasets, with 2000 configurations evaluated for each task. The design space contains architectural parameters (e.g. number of layers, for an

MLP) and training parameters (e.g. learning rate, dropout). For $h(\mathbf{x})$, we include the full training and validation accuracy and class-balanced accuracy curves. We evaluate methods on all 35 tasks in LCBench via cross-validation.

# 6. Results

For both domains in Sec. 5, we evaluate few-shot prediction of reward $f(\mathbf{x})$ on test tasks (the objective for the model during training), and optimization of test tasks.

**Baselines.** We compare our method to several baseline approaches. Existing transfer learning methods for BayesOpt do not make use of auxiliary information $h(\mathbf{x})$, however they provide an important comparison, indicating whether utilizing $h(\mathbf{x})$ with our approach yields measurable gains. We make a key comparison to an **Ours w/o h** baseline, which only provides $f(\mathbf{x})$ for the context points and uses an MLP to encode the context points in Fig. 2.

We also compare to a state-of-the-art transfer learning strategy for BO, which trains a shared GP on all the training tasks, parametrizing the learned kernel $K_\theta(\mathbf{x}, \mathbf{x}')$ with a neural network encoder (**DGP**, Wistuba & Grabocka (2021)). We additionally learn the mean function $\mu_\theta(\mathbf{x})$, which yielded improved performance (Wang et al., 2024). The GP with this learned prior is then deployed on test tasks.

We additionally implement a GP-based baseline which uses $h(\mathbf{x})$ (**GP-H**), to evaluate how effectively our model uses auxiliary information compared to other potential approaches. This baseline uses a multi-output GP which *jointly* models $h(\mathbf{x})$ and $f(\mathbf{x})$ by predicting the vector $(E_\theta(h(\mathbf{x})), f(\mathbf{x}))$, where $E_\theta$ is the same encoder of $h(\mathbf{x})$ as in our model. This GP is also trained across all the training tasks, and $E_\theta$ is learned jointly with the GP parameters.

Finally, we evaluate the standard single-task GP baseline (**STGP**), which fits a GP online to the data from the test task. Further details on all baselines are in Appendix E.2.

## 6.1. Gripper Design Task

**Implementation Details**. For training $P_\theta$, we sample the size of the context set in the range $[5, 30]$, and target set size is fixed at 100 examples. For the GP-based baselines, we sample 50 data-points and maximize the joint log-likelihood. See Appendix E for architecture and training details.

### 6.1.1. PREDICTION RESULTS

Since all methods build a surrogate model of the objective function, we compare our method against all baselines on few-shot prediction accuracy for target designs. We focus primarily on mean squared error (MSE) as the metric (summed over the examples in the target set), since we are focused on achieving accurate prediction of $f(\mathbf{x})$ for un-

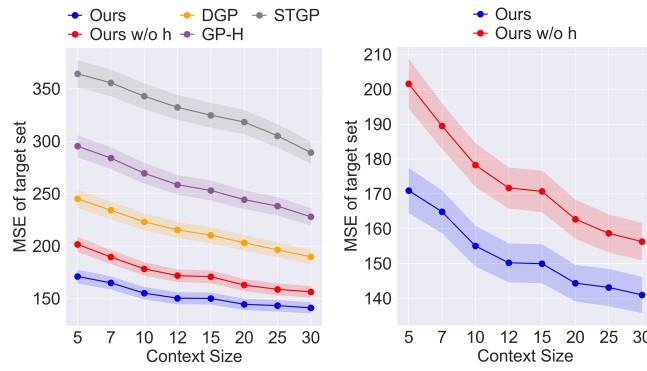

*(a)* Target MSE vs. context size.    *(b)* Comparison to Ours w/o h.

*Figure 4.* **Prediction results for gripper design**. Left shows the MSE on the target set for different context sizes, averaged across test tasks. Our method significantly outperforms all baselines, including SoTA baselines that utilize reward alone (DGP, Ours w/o h) and a GP-based approach for utilizing auxiliary information (GP-H). Right shows a direct comparison to Ours w/o h, highlighting our method's improvement especially for smaller context sizes. For each context size, 10 context/target set pairs were sampled for each test task (identically for all methods), and the MSE is averaged across all pairs and tasks. 95% confidence intervals are shown for each context size for all methods. Note that the MSE of each target set is summed over its 100 designs.

seen designs. Figure 4 shows our results, averaged over the test tasks. Our method demonstrates significantly improved few-shot prediction accuracy compared to all baseline methods, across a range of context sizes. While all methods that use a task history improve over the single-GP baseline, our method significantly outperforms SoTA baselines that utilize reward information alone (Ours w/o h, DGP). Notably, our method also significantly outperforms the GP-H baseline, indicating that ours better utilizes auxiliary information for prediction compared to a GP-based approach.

Figure 4b directly compares our method to the *Ours w/o h* baseline. Our method's significantly lower prediction error indicates the value of effectively utilizing auxiliary tactile information. Improvement is especially notable for smaller contexts: our model achieves an MSE of 170.9 vs. 201.6 for the baseline given a context size of 5 observations.

### 6.1.2. OPTIMIZATION RESULTS

We next evaluate our model on optimization of test tasks. For all experiments, we run discrete BayesOpt over the large set of designs evaluated for each task in our benchmark (on average 4300 designs). We perform 5 optimization runs for each test task with different initial contexts (chosen identically for all methods). Each initial context contains 5 designs, where no design has performance more than 30% of the maximum achievable reward. Each optimization run is performed for 30 trials. We use the Probability of Improvement acquisition function for all experiments.

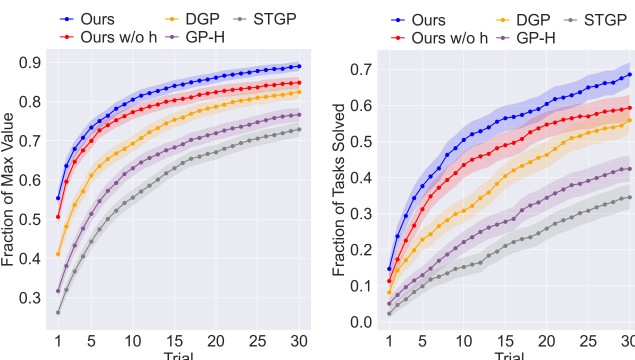

*(a) Avg. normalized reward*    *(b) Frac. Solved, as Regret ≤ 0.5.*

*Figure 5.* **Optimization of gripper design tasks**. (a) shows the best achieved value of $f$ (normalized by max $f$ for the task) over optimization, averaged across test tasks. Our model makes faster progress early on, and remains consistently better across optimization. In (b), we show that our method solves a significantly larger fraction of test tasks than all baseline methods by any number of trials of optimization. Average regret through optimization is shown in Appendix B, with the same conclusions. 95% confidence intervals across optimization runs are shown for all methods.

Results are shown in Fig. 5. Our method achieves significantly higher reward across the course of optimization compared to all baselines (Fig. 5a). These results indicate that our method can exploit auxiliary tactile information effectively to identify high-performing designs. Note again that our method outperforms all multi-task methods that utilize $f(\mathbf{x})$ alone, and the alternative GP-based approach for incorporating $h(\mathbf{x})$. After 30 trials, our model reaches almost 90% of the maximum achievable reward on average.

By 30 trials, our model achieves the maximum possible reward for 35.7% of test tasks, vs. 28.0% for the Ours w/o h model (the closest baseline). With a more conservative measure of solving a task (regret ≤ 0.5), our model solves 68.5% of test tasks vs. 59.3% for the closest baseline (Fig. 5b shows progress over optimization). Appendix A shows several qualitative examples of design tasks in the test set, demonstrating our method's ability to discover sophisticated grasping strategies such as clamping thin structures or protruding into gaps in the object to maximize grasp stability.

## 6.2. Hyperparameter Tuning Task

Methods are evaluated on all tasks in LCBench via cross-validation. Implementation details for all methods are similar to the gripper task, and are provided in Appendix E.

### 6.2.1. PREDICTION RESULTS

Figure 6 shows the results for few-shot prediction, averaged across all tasks. Our method demonstrates significantly improved prediction accuracy compared to all baselines, across a range of context sizes. While all multi-task meth-

ods outperform the single-GP baseline, the GP-H baseline performs relatively poorly. Our method also achieves noticeably lower prediction error than the other multi-task baselines, especially for smaller context sizes. Figure 6b provides a clear comparison to *Ours w/o h* in this regard. These results indicate that learning to use full per-epoch curves provides an advantage for few-shot prediction, notably even for a relatively smaller benchmark (34 training tasks, vs. several hundred).

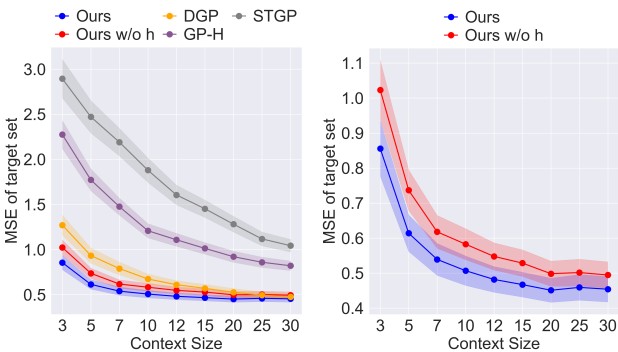

*(a) Target MSE vs. context size.*    *(b) Comparison to Ours w/o h.*

*Figure 6.* **Prediction results for hyperparameter design**. Left shows the MSE on the target set for different context sizes, averaged across tasks. Our method achieves lower prediction error than all baselines. While all multi-task methods outperform STGP, the GP-H baseline performs relatively poorly. Our method also achieves lower error than the other baselines, especially for smaller context sizes. The right figure shows this via a direct comparison to the Ours w/o h baseline. For each context size, 50 context/target pairs were sampled for each task (identically for all methods), and the results averaged across pairs and tasks. 95% confidence intervals are shown for each context size for all methods.

### 6.2.2. OPTIMIZATION RESULTS

For optimization, we curate a harder-to-solve subset of tasks in LCBench. Several tasks have identical optimal solutions, making optimization rather trivial; for instance, one HP configuration is the optimum for six tasks. Therefore, we curate the subset of tasks with unique optimal solutions ($\sim 50\%$ of tasks), presenting more of a challenge for optimization.

For each task, we perform 20 BayesOpt runs with an initial context size of 3 designs, and average results. Once again, the initial context is sampled from designs that achieved $\leq 30\%$ of the maximum achievable reward, and optimization is performed for 30 trials. The size of LCBench is most conducive to learning accurate point prediction over correct calibration, therefore we use a purely exploitative acquisition function for all experiments.

Results are shown in Figure 7. Our method achieves significantly higher reward throughout optimization compared to all baselines (Fig. 7a). By 30 trials, our method solves 89.7% of tasks (defined as within 0.01 of the maximum ac-

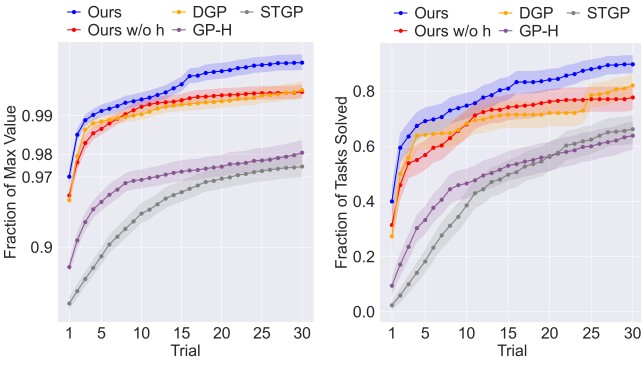

*(a)* Avg. normalized reward    *(b)* Frac. Solved (Regret $\leq 0.01$).

*Figure 7.* **Optimization on the LCBench benchmark**. (a) shows that our model achieves higher reward (normalized by maximum achievable $f$) over optimization. (b) shows that our method solves a significantly larger % of tasks by any point of optimization. The average regret is shown in Appendix B, with the same conclusions. 95% confidence intervals across optimization runs are shown for all methods.

curacy), compared to 82.1% for the closest baseline (Fig. 7b shows progress over optimization). These results indicate that our method utilizes learning curve feedback effectively to identify high-performing designs. Appendix D shows examples of optimization for two tasks, where our method quickly converges to optimal hyperparameter configurations, while the baselines either take far longer to converge or fail to converge entirely.

## 7. Discussion and Future Work.

Our results show that our approach, which learns to represent auxiliary information $h(\mathbf{x})$ through predicting the reward for unseen designs, enables more accurate prediction and more sample-efficient optimization of novel design tasks. Our approach outperforms current multi-task Bayesian optimization methods that utilize $f(\mathbf{x})$, demonstrating that measurable gains come from utilizing auxiliary feedback to solve design optimization tasks. It also outperforms an alternative GP-based approach for utilizing $h(\mathbf{x})$, suggesting that learning good representations of $h(\mathbf{x})$ for optimization is highly nontrivial. Our significant improvement on two challenging domains, robot gripper design and neural network hyperparameter tuning, indicates that our method transfers across diverse problem domains.

Our work opens the door to interesting directions of future work. For instance, in our method, $h(\mathbf{x})$ is not used as a prediction target; this is a deliberate choice as predicting the full content of $h(\mathbf{x})$ may distract from the objective of optimizing $f(\mathbf{x})$. However, there may be cases where predicting $h(\mathbf{x})$ offers a useful secondary learning objective, providing extra supervision. This may be more challenging

to do when $h(\mathbf{x})$ is very high-dimensional, such as in both domains we study. In this case, an encoder $E_\theta(h(\mathbf{x}))$ could first be learned via a self-supervised training objective. This pre-trained representation could then be inserted into the prediction model, fine-tuned for encoding $h(\mathbf{x})$ for context designs but frozen for predicting the encoded $h(\mathbf{x})$ of target designs. While we found that our end-to-end approach for learning a representation of $h(\mathbf{x})$ performed well, while being a more general approach, this two-stage pipeline could be explored in future work.

In our work, the test task is assumed to come from the same distribution as the training tasks. In real-world situations it is possible a system could encounter an extremely out-of-distribution task; while the task can be folded into the model post-hoc, it must be solved first to a reasonable degree. For such tasks, one approach is to ensemble the trained model $P_\theta$ with a 'cold-start' model fit only on test-time data. If $P_\theta$ displays high prediction error or high uncertainty estimates during optimization, weight could shift to the cold-start model for choosing new observations. We see the problem of recognizing and handling highly-OOD tasks as an interesting direction for future methods development.

Finally, we note that our design optimization method aims to maximize the immediate gain in reward at each iteration. This is the standard one-step look-ahead approach for BayesOpt, which generally proves successful. However, in some problems, it might be helpful to employ a more 'long-horizon' strategy, where designs can be selected for the purpose of deeper *understanding* of the task, which then enables finding a higher-reward design at a future optimization trial. The presence of auxiliary information, which provides much richer task information during each trial, makes such a strategy potentially relevant to our setting, which is an interesting direction to explore and assess in future work.

## Dataset Release

The gripper design benchmark is available to download on the paper website, designopt.cs.columbia.edu.

## Acknowledgments

We thank Huy Ha for helpful advice on the MuJoCo simulator. This work is supported by the funds provided by the National Science Foundation and by DoD OUSD (R&E) under Cooperative Agreement PHY-2229929 (The NSF AI Institute for Artificial and Natural Intelligence). Arjun Mani is supported by the NSF Graduate Research Fellowship.

## Impact Statement

Our work introduces a new design optimization setting, motivated by real-world experimental setups, and our pro-

posed method demonstrates improvements in design optimization. Our work has applications to design in different scientific and engineering domains (such as robotics, drug design), and holds promise for contributing to advances in these domains. We thus expect our work to contribute to a positive societal impact. Like any design optimization method, the possibility that our method could also be applied towards more harmful purposes cannot be excluded completely. However, we believe that the positive applications of our work strongly outweigh that risk.

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

## A. Qualitative Examples of Gripper Design Tasks.

Figure 8 shows examples of two design tasks in the test set, comparing our method against the *Ours w/o h* baseline. In the first example, at step 0, the gripper picks up a bottle by a flat *friction* grasp, which cannot withstand much disturbance. By the end of optimization, our method discovers a stable grasp that curves around the bottle to support its sides, while supporting its base from below. By contrast, the $f$-only model is unable to improve from the initial context; presumably, our method can leverage the rich tactile feedback available even in grasp *failures* to infer an appropriate gripper. For the second example of an airplane object, the initial gripper fails to pick up the airplane (shown from a top-down view). By Step 30, our model finds a stable grasp that on first contact rotates the airplane 45 degrees, pushing one wing in front and one wing behind the gripper. This suggests that the model learns to leverage *dynamics* for stable grasps, not just object shape. In contrast, the baseline model's final grasp can be dislodged by applying a force to the nose. These examples indicate our model's potential for finding creative solutions to design problems.

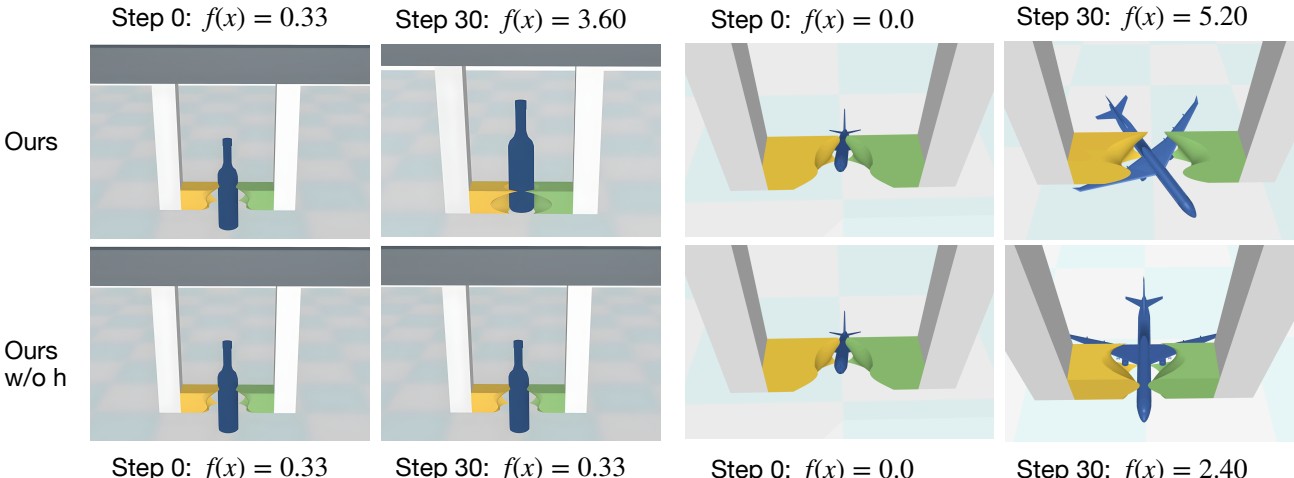

*Figure 8.* **Comparison to the *Ours w/o h* baseline on two design tasks.** For the left task, the gripper design at Step 0 cannot resist much disturbance. By Step 30 of optimization, our model finds a stable gripper that curves around the bottle, while the baseline does not improve from the initial context. For the right task, the gripper at Step 0 fails to pick up the airplane (a top-down view is shown). Our model's solution rotates the airplane to push one wing in front and one behind the gripper, while supporting it from below, resisting disturbances in all three force directions. This solution achieves the maximum possible reward. By contrast, the baseline's final grasp can be dislodged by applying a force to the nose.

Figure 9 shows **several more qualitative examples of design tasks** in the test set, as well as a 2D tSNE visualization of a large subset of the test set. For each design task, the ground-truth optimal gripper is shown, as well as the gripper design discovered by our method during optimization. Our model discovers sophisticated grasping strategies that maximize grasp stability, such as clasping the seat of a chair to resist disturbances in any vertical or horizontal direction (first row), clamping thin structures like the legs of a piano (second row), and protruding into any openings in the object structure (third row).

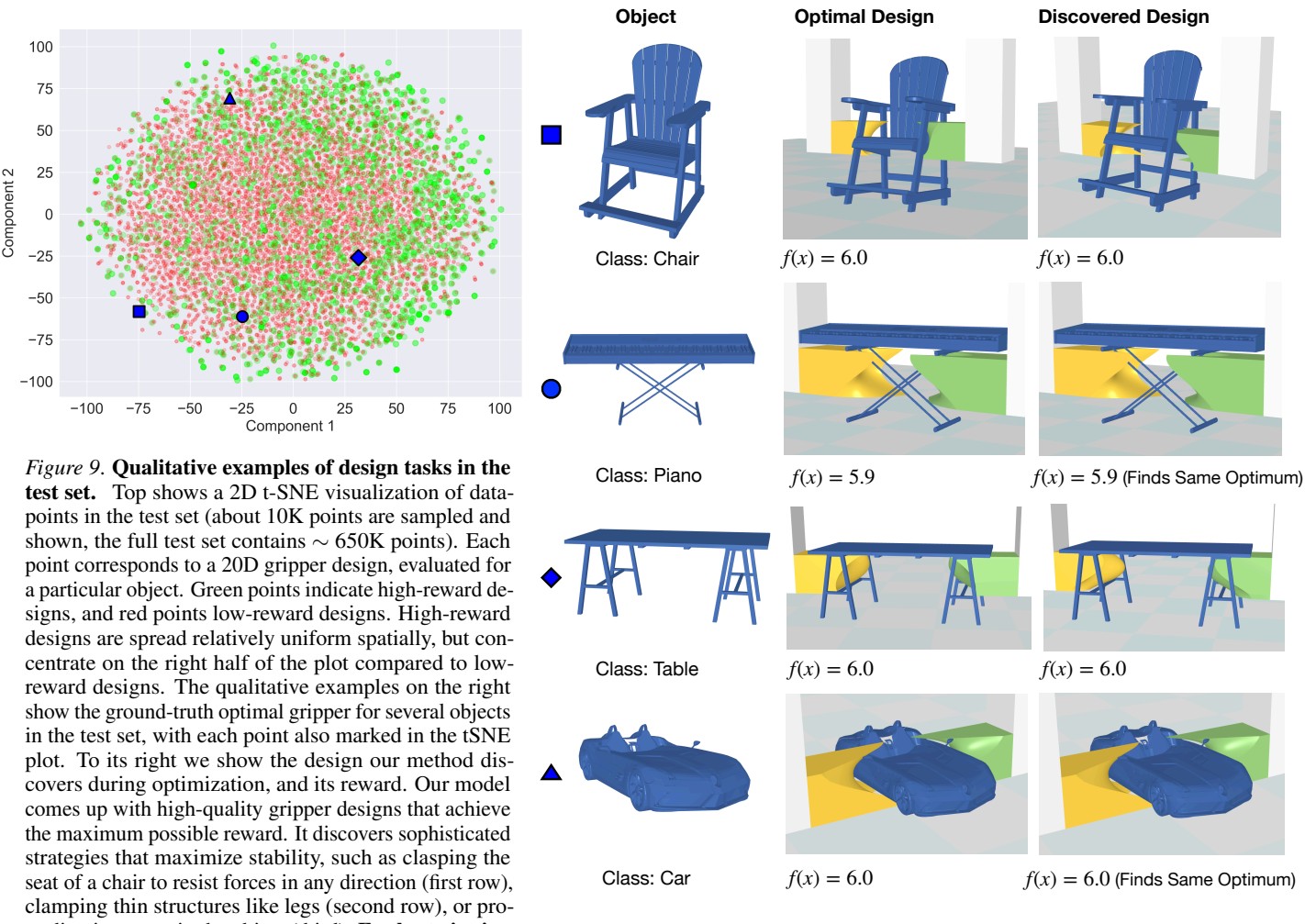

*Figure 9.* **Qualitative examples of design tasks in the test set.** Top shows a 2D t-SNE visualization of datapoints in the test set (about 10K points are sampled and shown, the full test set contains ∼ 650K points). Each point corresponds to a 20D gripper design, evaluated for a particular object. Green points indicate high-reward designs, and red points low-reward designs. High-reward designs are spread relatively uniform spatially, but concentrate on the right half of the plot compared to low-reward designs. The qualitative examples on the right show the ground-truth optimal gripper for several objects in the test set, with each point also marked in the tSNE plot. To its right we show the design our method discovers during optimization, and its reward. Our model comes up with high-quality gripper designs that achieve the maximum possible reward. It discovers sophisticated strategies that maximize stability, such as clasping the seat of a chair to resist forces in any direction (first row), clamping thin structures like legs (second row), or protruding into gaps in the object (third). **For best viewing of the gripper designs**, please zoom into each row.

## B. Optimization Results for Average Regret.

We provide the average regret during optimization for both domains, along with the other results in the main text. Figure 10 shows the results for the gripper design task, and Figure 11 shows the results for the hyperparameter design task.

## C. Pseudocode of Training and BayesOpt procedures.

Figure 12 shows pseudocode for our training procedure and Bayesian Optimization procedure.

## D. Qualitative Examples for the Hyperparameter Tuning Benchmark.

Figure 13 shows qualitative examples of optimization for two tasks in LCBench, comparing our method against the two strongest baselines (Ours w/o h, DGP). For both tasks, our method finds a hyperparameter configuration that achieves the maximum possible reward, unlike the baseline methods. The figure also shows the evolution of two hyperparameters during optimization, the learning rate and the number of hidden units in the first layer (which determines all layers, since the architecture is a funnel-shaped MLP). In the first task, our method converges to a lower learning rate and a smaller network, matching the optimal configuration, while both baselines remain stuck at higher hyperparameter values. In the second task, our method again converges to optimal values of the two HPs within 20 trials, while the baselines oscillate significantly and take longer to converge. These examples indicate our method's ability to either solve design tasks more efficiently (second task), or solve tasks where other methods fail entirely (first task), by effectively leveraging auxiliary feedback.

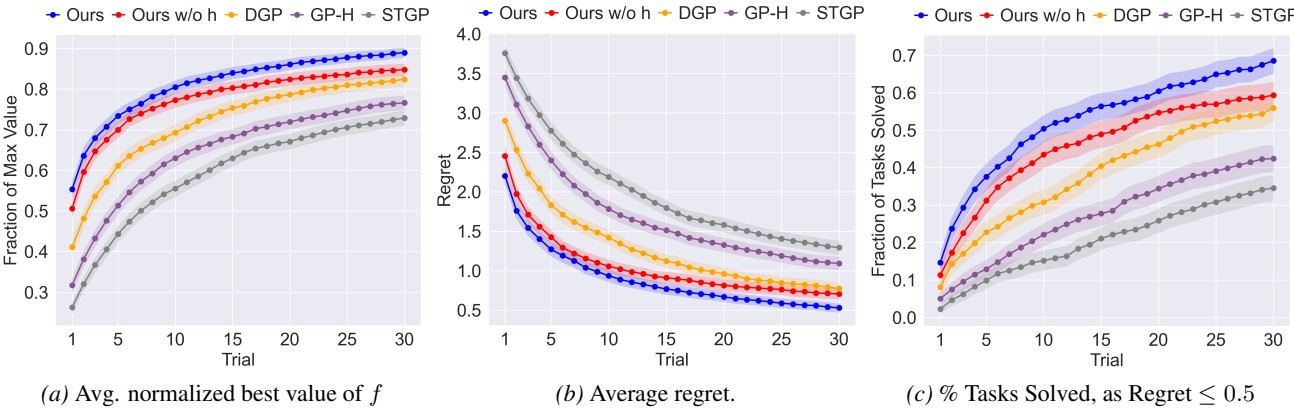

*(a) Avg. normalized best value of $f$*    *(b) Average regret.*    *(c) % Tasks Solved, as Regret $\leq 0.5$*

*Figure 10.* **Optimization of gripper design tasks**. (a) shows the best achieved value of $f$ (normalized by max $f$ for the task) over optimization, averaged across test tasks. (b) shows the average regret. Both (a) and (b) show that our model makes faster progress early on, and remains consistently better across optimization. In (c), we show that our method solves a significantly larger % of test tasks than all baseline methods by any number of trials of optimization.

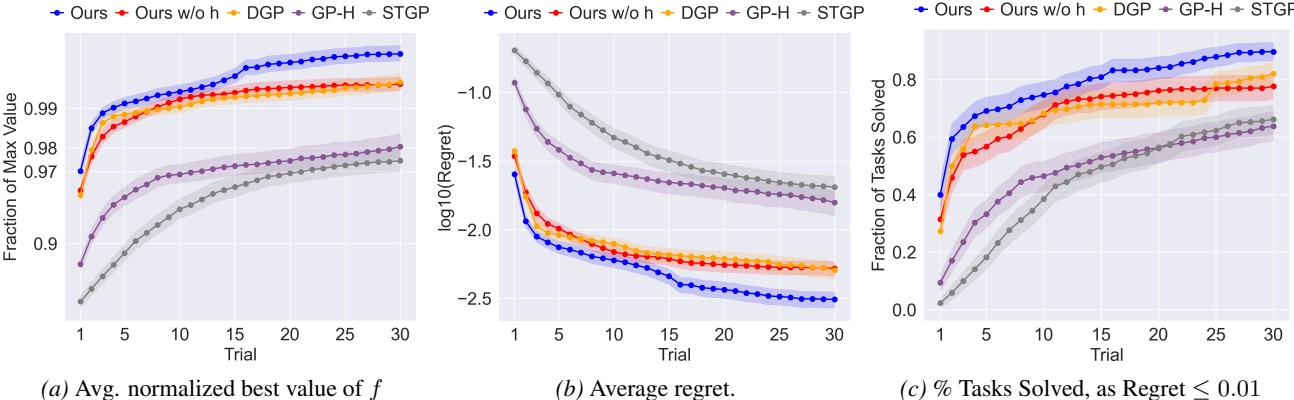

*(a) Avg. normalized best value of $f$*    *(b) Average regret.*    *(c) % Tasks Solved, as Regret $\leq 0.01$*

*Figure 11.* **Optimization on the LCBench benchmark**. (a) shows that our model achieves higher reward (normalized by maximum achievable $f$) over optimization, averaged across tasks, and (b) shows our model's lower average regret. In (c), we show that our method solves a significantly larger % of tasks by any point of optimization.

## E. Architecture and Training Details for All Methods.

### E.1. Our method implementation details.

**Gripper Design Task**. For training, we sample context set size uniformly in the range $[5, 30]$, and the target set size is fixed at 100 examples. For both context and target sets, we sample low-reward and high-reward data-points with equal probability, ensuring that the model has an 'informative' context and also learns to accurately predict high-performing designs.

For the model architecture, we use a model dimension of 256 and 9 layers for both the context encoder transformer and predictor transformer in Fig. 2. The target encoder is a one-layer MLP. For the context encoder, since three simulations are run to evaluate each gripper design, $h(\mathbf{x})$ actually consists of three sequences of tactile observations (this is described further in Appendix G). We concatenate the information from all three sequences at each time-step, such that a single sequence is provided to the context encoder in Fig. 2b. For each time-step, the two tactile images from each of the three sequences, of shape $2 \times 16 \times 16$, are encoded by a CNN; the image encodings for each sequence are then concatenated and further concatenated to the other features for all the sequences. This concatenated vector is passed into an MLP which produces a final embedding for that time-step. This accounts for the per-timestep input encoder which goes into the transformer in Fig. 2b. The rest of the context encoder architecture is described sufficiently in Sec. 4.2.

We train $P_\theta$ with the AdamW optimizer, using a learning rate of 1e-4, a batch size of 8 tasks, weight decay of 0.01, and dropout of 0.2. We monitor performance on the validation set (75 design tasks) throughout training, and use early stopping.

**Algorithm 1** Training Procedure

**Input:** Set of Training Tasks $\mathcal{T}_{train} = \{\mathcal{T}^{(1)}, ... \mathcal{T}^{(N)}\}$, each with an associated dataset $\mathcal{D}^{(i)} = \{(\mathbf{x}_j^{(i)}, F^{(i)}(\mathbf{x}_j^{(i)}))\}_{j=1}^{n^{(i)}}$, initialized model $P_\theta$.

**for** $t = 1,...$ **do**
    **for** *task* $\mathcal{T}^{(i)}$ *in* $\mathcal{T}_{train}$ **do**
        Sample an integer $N_C < |\mathcal{D}^{(i)}|$.
        Sample an integer $N_T \leq |\mathcal{D}^{(i)}| - N_C$.
        Sample $C \subset \mathcal{D}^{(i)}$, s.t. $|C| = N_C$.
        Sample $T \subset \mathcal{D}^{(i)} \setminus C$, s.t. $|T| = N_T$.
        For $\mathbf{x}_k \in T_\mathbf{x}$, predict $\hat{f}^{(i)}(\mathbf{x}_k) = \mathcal{N}(\hat{\mu}_k, \hat{\sigma}_k)$, using the model $P_\theta(\cdot|C, T_\mathbf{x})$ (Fig. 2).
        Compute $\mathcal{L} = \sum_k \log \mathcal{L}_k$, where $\mathcal{L}_k$ is the likelihood of $f^{(i)}(\mathbf{x}_k)$ under $\hat{f}^{(i)}(\mathbf{x}_k)$.
        Calculate $\nabla_\theta \mathcal{L}$, update model weights $\theta$.
    **end**
**end**

*(a)* Training procedure

**Algorithm 2** Bayesian Optimization

**Input:** Initial observations for test task $C_0 = \{(\mathbf{x}_1, F(\mathbf{x}_1)), ..., (\mathbf{x}_{n_0}, F(\mathbf{x}_{n_0}))\}$, trained model $P_\theta$, search space $\mathcal{X}$, number of BO iterations $K$, acquisition function $\alpha$.

**for** $t \leftarrow 0$ **to** $K - 1$ **do**
    Set $\mathcal{X}_t \leftarrow \mathcal{X} \setminus C_{t_\mathbf{x}}$.
    Find $\mathbf{x}_{t+1} \in \arg\max_{\mathbf{x} \in \mathcal{X}_t} \alpha(\mathbf{x}, P_\theta(\cdot|C_t))$, using the predictions $\hat{f}(\mathbf{x}) = P_\theta(\cdot|C_t, \mathbf{x})$.
    Observe $F(\mathbf{x}_{t+1}) = (f(\mathbf{x}_{t+1}), h(\mathbf{x}_{t+1}))$.
    Update the context with the new observation $C_{t+1} \leftarrow C_t \cup (\mathbf{x}_{t+1}, F(\mathbf{x}_{t+1}))$.
**end**
**return** best configuration: $\arg\max_{(\mathbf{x}, F(\mathbf{x})) \in C_K} f(\mathbf{x})$.

*(b)* Bayesian Optimization procedure

*Figure 12.* **Pseudocode for our training procedure and BayesOpt procedure.** For any set $S$ of ordered pairs, $S_\mathbf{x}$ denotes the restriction of $S$ to the inputs $\mathbf{x}$ only. In practice, for (a), we sample a batch of tasks at each iteration of the inner loop. The pseudocode for (b) is agnostic to whether $\mathcal{X}$ is discrete or continuous, the only difference being how optimization of $\alpha$ is conducted at each iteration.

**Hyperparameter tuning task**. The context set size is sampled uniformly in the range [3, 30], and the target set size is fixed at 100 examples. For the target set, we sample low-reward and high-reward data-points with equal probability, ensuring that the model learns to predict high-performing designs. The context set is sampled uniformly. Our method (and all baselines) is evaluated on all tasks in LCBench via cross-validation: each task is held out and the remaining tasks are used for training.

For the model architecture, we use a model dimension of 256 and 6 layers for both the context encoder transformer and predictor transformer in Fig. 2. The target encoder is a one-layer MLP. Note that $h(\mathbf{x})$ consists of five per-epoch curves over 50 epochs, i.e. $h(\mathbf{x}) \in \mathbb{R}^{50 \times 5}$. These are the train and validation accuracy and class-balanced accuracy curves, as well as the learning rate at each epoch (which follows a cosine annealing schedule). Therefore, each time-step of $h(\mathbf{x})$ is a 5-dimensional vector, and is encoded with an MLP in the context encoder.

We train $P_\theta$ with AdamW with a learning rate of 1e-4, a batch size of 3, weight decay of 0.01, and dropout of 0.1. Note that we evaluate on LCBench via cross-validation; each task is held-out as a test task, and the remaining tasks are used for training $P_\theta$. We train each model for 5000 epochs.

### E.2. Baselines Implementation Details.

#### E.2.1. OURS W/O H

This model omits $h(\mathbf{x})$ from the context points in Fig. 2. A one-layer MLP is used for both the context and target encoders. The transformer model dimension and number of layers are the same as for our model in each domain. Training protocol (context set sampling, target set sampling) and hyperparameters (learning rate, dropout, weight decay, batch size) are also consistent for our model and this model.

#### E.2.2. DGP

This model learns a shared GP across the training tasks. The kernel takes the form $K_\phi(E_\theta(\mathbf{x}), E_\theta(\mathbf{x}'))$, where $E_\theta$ is a neural network encoder and $\phi$ are the kernel hyperparameters. For both domains, $E_\theta$ is an MLP with three hidden layers. For the kernel $K_\phi$, we use either an RBF kernel or a Matern-5/2 kernel, both with ARD. The embedding dimension (i.e. the output

**Task 1** (LCBench dataset, 4-way classification)

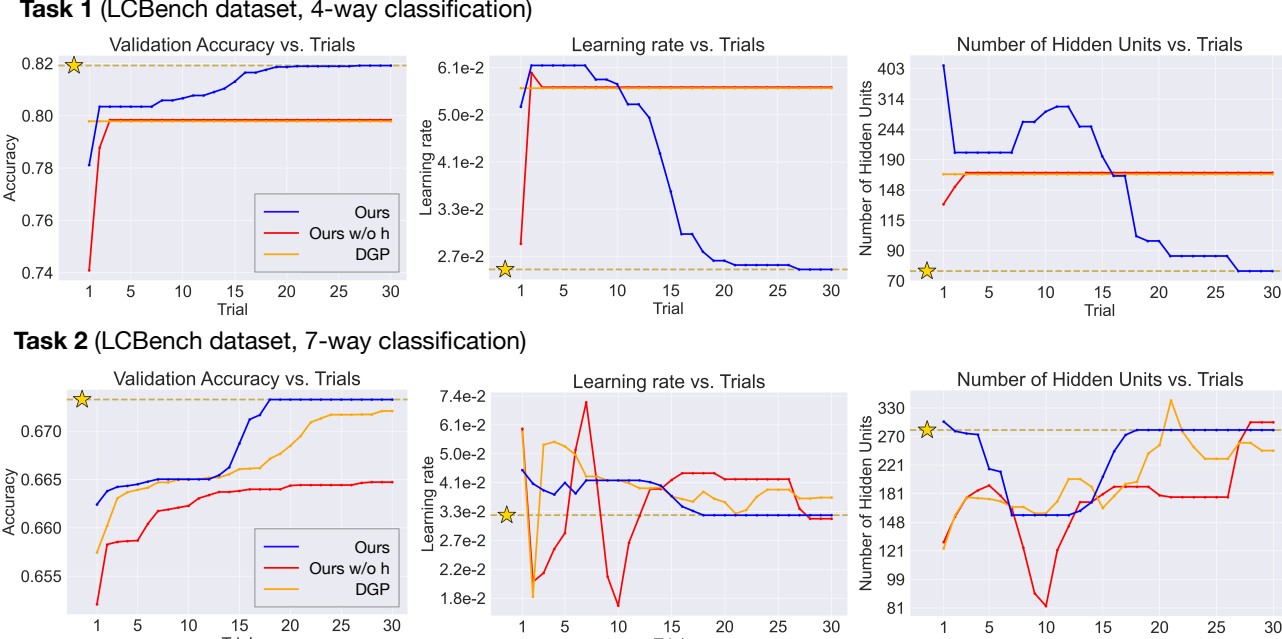

**Task 2** (LCBench dataset, 7-way classification)

*Figure 13.* **Examples of two hyperparameter tuning tasks in LCBench**. The top row shows a 4-way classification task, and the bottom row a 7-way classification task. The left column shows how the reward $f(\mathbf{x})$ evolves over optimization for our method and the two strongest baselines (Ours w/o h, DGP). For both tasks, our method converges to the maximum achievable accuracy (marked with a star), while the baselines either make little to no optimization progress (Task 1) or converge less quickly and satisfactorily (Task 2). The right columns shows the evolution of two hyperparameters, the learning rate and the number of units in the first hidden layer of a funnel-shaped MLP. The optimal value of each hyperparameter is indicated. In the first task, our method converges to a lower learning rate and smaller network size, matching the optimal configuration, while the baseline methods remain stuck on higher hyperparameter values. In the second task, our method converges to optimal HP values within 20 trials, while the baselines display significant oscillation. Note that at every trial, the best observed reward and configuration so far are shown, hence why the line for some methods are flat.

dimension of $E_\theta$), as well as the choice of kernel, are chosen via grid-search on the validation set. We additionally learn the mean function $\mu_\theta$, as proposed by (Wang et al., 2024), which we found improved performance on both domains. As proposed by the authors, we parametrize the mean function as a learnable linear layer on top of the encoder, i.e. $\mu_\theta(\mathbf{x}) = WE_\theta(\mathbf{x})$.

As suggested by previous work, for both domains, we sample 50 data-points from the training task at each iteration and maximize the joint log-likelihood of the observations. This set is equally balanced between low-reward and high-reward observations. For both domains, we train with AdamW with a learning rate of 1e-4, and weight decay of 0.01. Batch size is consistent with that used for our model in either domain. This model and the models below are implemented in GPytorch (Gardner et al., 2018).

### E.2.3. GP-H

This baseline trains a multi-output GP that outputs the vector $(E_\theta(h(\mathbf{x})), f(\mathbf{x}))$, where $E_\theta$ is a neural network encoder of $h(\mathbf{x})$. The encoder $E_\theta$ is the same encoder of $h(\mathbf{x})$ as shown in Fig. 2, that produces $e^{seq}$; however in this case, this is *not* added to an embedding of $(\mathbf{x}, f(\mathbf{x}))$. For the multi-output GP, we use the standard ICM kernel, which factors the kernel $K((\mathbf{x}, y), (\mathbf{x}', y'))$ into a kernel over data-points $K(\mathbf{x}, \mathbf{x}')$ and a correlation matrix over output vector elements $K(y, y')$. This matrix in particular enables the GP to model how $h(\mathbf{x})$ correlates with and informs the prediction of $f(\mathbf{x})$. For maximum expressivity, we use a full-rank correlation matrix for $K(y, y')$. For the kernel $K(\mathbf{x}, \mathbf{x}')$, we use either an RBF or Matern-5/2 kernel with ARD. The output dimension of $E_\theta$, which determines the number of outputs for the GP, as well as the kernel type, are chosen via grid-search on the validation set.

As for the previous baseline, we sample 50 points from the training task at each iteration. The encoder $E_\theta$ is first applied to the observations of $h(\mathbf{x})$ in these points. Then, the joint likelihood of the *multi-output* observations $(E_\theta(h(\mathbf{x})), f(\mathbf{x}))$ is maximized. The encoder is learned jointly with the GP parameters to maximize this objective; ideally this allows the model to learn how to represent $h(\mathbf{x})$ such that it correlates strongly with $f(\mathbf{x})$ and improves prediction. We train with AdamW with a learning rate of 1e-4 and weight decay of 0.01. Batch size is consistent with that used for our model in either domain.

### E.2.4. STGP

The single-task GP baseline is fit only on the test-time context. Each time the context is updated, the model is refit to the new context, contrasting with *all* multi-task methods that learn representations from the training tasks and are then frozen at test-time. We use the state-of-the-art kernel in BoTorch (Balandat et al., 2020), which is an RBF kernel with a log-normal prior over the lengthscale that scales with input dimensionality.

## F. Size Comparison of our Benchmark and Existing Multi-Task BayesOpt Benchmarks.

Our gripper design benchmark is a large-scale benchmark, consisting of 4.3 million gripper evaluations across 997 design tasks. Generating the benchmark involved *1.07 million* CPU hours of simulation. Notably, our benchmark is significantly larger than nearly all multi-task BayesOpt benchmarks. The only comparable benchmark in the literature is HPO-B (Pineda-Arango et al., 2021), with 6.39 million evaluations over 196 tasks. Crucially however, these tasks have different input search spaces, making transfer learning very difficult; thus in practice, much smaller subsets with a common search space are used for benchmarking. Our benchmark has a common design space. Other than this, popular benchmarks collected in the HPOBench suite (Eggensperger et al., 2021) have at least one order of magnitude less data-points, and benchmarks in other domains are several orders smaller (Maraval et al., 2023). To our knowledge, our benchmark contains many more tasks (997) than previous benchmarks ($< 200$), enabling learning richer representations for optimization. These benchmarks also generally do not provide auxiliary information, unlike ours.

## G. Implementation Details for Gripper Design Benchmark

We provide further details of the gripper design task below.

**Simulation.** The gripper simulation consists of a wrist with an up-down position-based actuator, and left and right handles with force-based actuators. The left and right grippers close in on the object with 1.0N of force for 250 time-steps, at which point it has settled into a grasping pattern. Then the object is lifted to a height of 0.12 m. Once the object has stabilized, disturbance forces are applied in a fixed direction starting at $F = 0.5$N and incrementing by 0.1N. Each force is applied for 20 time-steps; after each force application, the object is allowed to re-stabilize before incrementing the force. The simulation runs for a maximum of 5000 time-steps, terminating early if the object falls from the grasp and touches the ground. The default time-step of 0.002 seconds in MuJoCo is used. The object has a fixed mass of 0.05 kg, and each gripper has a fixed mass of 0.1 kg. For contact modeling which requires convex geometries, we perform convex decomposition of both the object and the gripper using the CoACD library before running the simulation (Wei et al., 2022).

Three simulations are run with different force directions: $(-z)$ (down), $(-x)$ (into the page), and $(+x)$ (out of the page). The reward for each simulation is the maximum disturbance force applied before the object fell out of the grasp, or at the end of the simulation (maximum achievable reward is around 6.0). This indicates how well the grasp was able to resist disturbances in a particular direction. The final reward for a design is this metric averaged over the three simulations $f(\mathbf{x}) = (f_1(\mathbf{x}) + f_2(\mathbf{x}) + f_3(\mathbf{x}))/3$, as a measure of robustness to perturbations in different directions.

**Tactile Feedback.** The inward face of each gripper is divided into a $16 \times 16$ uniform tactile grid. At each time-step, MuJoCo computes contact points $(\mathbf{p}_i, \mathbf{n}_i, \mathbf{F}_i)$, where $\mathbf{p}_i$ is the position of the contact, $\mathbf{n}_i$ is the contact normal, and $\mathbf{F}_i$ is the contact force in the local frame. Like standard approaches (e.g. MuJoCo's touch sensor), we record the contact force in the normal direction. The contact point is mapped to a 'taxel' in the grid based on its location (this often requires projecting the point to the gripper surface via the contact normal). If the contact point is on another face than the inward face (e.g. the gripper is supporting the object from its top face), a scalar contact reading is recorded for that face.

The result of this process is the following information at each time-step: two $16 \times 16$ tactile images $\text{Ti}_L$ and $\text{Ti}_R$ for the inward face of each gripper, and two 5-dimensional vectors $\text{Tf}_L$ and $\text{Tf}_R$ for the scalar contact readings on the other faces (top face, bottom face, back face, front face, and handle). Finally, we include a small amount of state information $q$ at each time-step, including the position and velocity of all the joints (6-dim), and the disturbance force applied at that time-step, which could be 0. The two tactile images, two face readings, and system state becomes the information at each time-step. This information is recorded at each time-step in the simulation. Since there are three simulations, the tactile sequence is recorded for each simulation and $h(\mathbf{x})$ contains the three sequences $h(\mathbf{x}) = (h_1(\mathbf{x}), h_2(\mathbf{x}), h_3(\mathbf{x}))$. Notably, the features at each time-step are heterogeneous, and the tactile information at each time-step tends to be quite sparse; thus capturing the insights present in this complex observation sequence effectively for prediction is a challenging task.

**Data Generation.** A total of 997 objects are sampled from ShapeNet, proportionally to class frequency in the dataset. Each object is normalized to fit within a bounding box of $(9 \text{ cm})^3$, and rotated so that its symmetry axis is the $x - z$ plane and it appears symmetrically to both grippers. The gripper fingers are each of size $(2 \text{ cm})^3$. For each object, 400 gripper geometries are sampled from the Bezier curve parametrization described in Sec. 5, and each gripper geometry is evaluated at increments of $0.5$ cm from a height of $0.0$ to until the gripper base has exceeded the height of the object. This leads to $400 * H$ gripper designs evaluated for each object, depending on its height $H$; the maximum possible number of evaluations is 7200, and the general range is from 1000 to 7200.

**Dataset Statistics.** There are a total of 4,278,400 data points in the dataset, across 997 unique objects. This evaluates to about 4290 designs per object on average, although it varies according to the range above. 31.3% of objects have a design with 6.0 reward (the highest achievable reward), 65.3% have a design with at least 5.0 reward, and 78.5% have a design with at least 4.0 reward. This indicates that most objects have a high-performing design for grasping, and the design problem is to find these high-quality designs. Only 2.7% of objects have a maximum reward of less than 1.0, a very small percentage corresponding to objects with no legitimate grasping strategy (such as a cabinet with completely flat sides, essentially a box). For each object, lower-performing designs are much more frequent than high-performing designs, as is expected for a complex task of this nature. On average, a task contains 65.9% of designs with zero reward and about 4.4% of designs that are within 20% of the maximum reward. This creates a challenging optimization problem to find high-performing designs for a task, given that most candidates do not lead to successful grasps. Correspondingly it implies that a method that does succeed in this task is capable of addressing complex design problems.

