# OpenReview forum: "Few-Shot Design Optimization by Exploiting Auxiliary Information"
_ICML.cc/2026/Conference — ICML 2026 regular_

### Official Review · Reviewer_iLyv · 2026-02-24

**Soundness:** 3
**Presentation:** 2
**Significance:** 3
**Originality:** 2
**Overall Recommendation:** 5
**Confidence:** 4

**Summary:**

This paper studies few-shot black-box design optimization where each evaluation of a design $x$ returns both a scalar objective $f(x)$ (the quantity to optimize) and auxiliary information $h(x)$ (additional feedback that helps explain why $f(x)$ is high or low). The authors propose a meta-learned surrogate model that conditions predictions of $f(x)$ for new candidate designs on a small context set of previously evaluated designs, using $h(x)$ from the context to improve data efficiency. The model combines an encoder for $h(x)$ (for high-dimensional or sequential feedback) with a transformer-based set model to produce a predictive distribution over rewards, and is then used inside a Bayesian optimization loop. Experiments across multiple domains suggest that leveraging auxiliary information improves few-shot prediction and optimization performance compared to baselines that rely only on $(x,f(x))$ observations.

**Compliance With Llm Reviewing Policy:**

Affirmed.

**Final Justification:**

This is a clear, technically solid, and practically meaningful paper on few-shot design optimization with auxiliary feedback. The rebuttal addressed my main concerns about evaluation fairness, statistical reporting, and practical overhead, which strengthened my confidence, so I remain supportive of acceptance.

**Key Questions For Authors:**

I enjoyed reading the paper---thanks for putting this work together.

1. Could you clarify the BO initialization and randomness protocol for both Gripper and LCBench? Specifically: (i) how the initial context designs are sampled (beyond the “$\leq 30\\%$ of max” constraint for Gripper), (ii) what sources of randomness/seeds are used across repeated runs, and (iii) whether all methods/baselines are evaluated with the exact same initial contexts (and the same candidate pools) for each run/task to ensure a fair comparison?

2. For LCBench, could you specify the exact acquisition function used (formula/implementation)?

4. Could you please include the error bars (e.g., $95\\%$ confidence intervals or mean$\pm$std) on the plots across runs/tasks.

5. Since the setting (leveraging auxiliary information per evaluation) can be easily conflated with multi-fidelity (MF) or multi-information source (MIS) BO, could you add a short (1 or 2 lines) clarification on how $h(x)$ here differs from a fidelity/source and how the proposed approach relates to (or differs from) MF/MIS BO formulations?

**Limitations:**

I understand the page limit constraints, but a few additional details would be helpful:

1. Adding a note on robustness when $h(x)$is uninformative/misaligned with $f(x)$ (since the method’s benefit depends on that relationship).

2. Short discussion of practical overhead / reliability of collecting $h(x)$ in real systems (sensor drift, synchronization, compute), and how that might affect deployment.

**Strengths And Weaknesses:**

## Strengths

- **Well-motivated problem setting.** The paper targets a realistic design-optimization setting where each evaluation yields not only a scalar reward $f(x)$ but also auxiliary information $h(x)$ (e.g., tactile trajectories / learning curves), which is common in many domain including robotics, AutoML, etc.

- **Clean conceptual framing of auxiliary information.** Treating $h(x)$ as *information to condition on* (to improve prediction and optimization of $f(x)$) avoids awkward multi-objective formulations and leverages feedback without requiring hand-designed scalarizations of $h(x)$.

- **Meta-learning across related tasks.** Learning a task-distribution-level surrogate that can rapidly adapt from a small context is a strong fit for few-shot BO, and aligns well with practical scenarios where many related tasks are available.

- **Modular modeling.** The approach cleanly separates (i) encoding $h(x)$ and (ii) a set-conditioned predictive model for $f(x)$, making the framework adaptable to different forms of auxiliary data and domains.

- **Clear writing:** The paper is easy to read, and the method description is mostly straightforward to follow.

## Weaknesses

- **When auxiliary feedback is uninformative or noisy.** The apporach assumes $h(x)$ is meaningfully predictive of $f(x)$. If $h(x)$ is noisy or weakly correlated, the benifit may dminish or even become negative.

- **Auxiliary information collection/processing cost.** While $h(x)$ can improve sample efficiency, the method implicitly assumes auxiliary trajectories can be reliably captured, synchronized, stored, and encoded. In real robotic pipelines, this overhead (and potential sensor drift/misalignment) can materially affect throughput and robustness, so it would help to discuss compute/logging requirements or provide an ablation with compressed/cheap $h(x)$. [Please see the `Detailed Comments` section]

- **Evaluation setting may be narrower than general design BO.** In the gripper study the optimization appears to operate over a finite candidate pool of pre-generated designs. it’s unclear how well results transfer to truly continuous design spaces where acquisition optimization is harder.

- **Experimental reporting.** As far as I can tell, the paper does not report variance/error bars on the plots and omits key BO protocol details (initial context sampling, seed control, and whether initial contexts are matched across methods), which makes it harder to assess robustness and fairness. [this is the main reason for the “fair” presentation rating.]

## Detailed Comments
### Regarding auxiliary information collection/processing cost:
  - The paper motivates "few-shot optimization" where evaluations are expensive. But in some real robotics settings preprocessing + encoding $h(x)$ (such as the long tactile sequences) can be become a $\textit{comparable}$ or $\textit{dominat}$ cost. That changes whether the method is actually “sample-efficient” in wall-clock / engineering terms.

- Informative $h(x)$ in robotics often requires reliable time synchronization across multiple ROS topics, careful sensor calibration, and nontrivial preprocessing (e.g., filtering, alignment, and downsampling). In practice, small changes in sensor setup, latency, or drift can shift the distribution of $h(x)$, potentially degrading the learned encoder and reducing performance at deployment.

- If baselines don’t use $h(x)$, they also don’t pay its compute/engineering cost. If the paper claims “better efficiency," it’s fair to ask: efficiency in $\textit{sample count}$ only, or also in $\textit{compute/latency}$?

### Originality
In my view, the overall idea feels more like a nice combination of known pieces (using auxiliary information, meta-learning across tasks, transformer encoders) than a fundamentally new concept. The main contribution is the framing and showing it works in these settings, which is totally fine, but it’s less of a brand-new algorithmic idea.

### A few caveats (not necessarily weaknesses)
- **Uncertainty in the acquisition:** The BO step uses Probability of Improvement, which depends on the predicted standard deviation $\sigma$. Since $\sigma$ comes from a deterministic NLL-trained model (i.e., negative log-likelihood term $-\log P_{\theta} (\cdot)$), it may not reliably indicate where the model is uncertain, which could affect exploration behavior. A brief note clarifying whether $\sigma$ is treated mainly as a heuristic for PI (rather than a calibrated uncertainty estimate) would make the setup easier to interpret.

- **Weaker inductive bias than kernel-based BO:** Attention-based conditioning does not enforce locality/smoothness structure in $x$-space (unlike kernel-based surrogates), so generalization may be less stable unless the training task/data distribution closely matches test-time behavior.

---

> ### Author Rebuttal · Authors · 2026-03-29
>
> Thank you for your detailed and thoughtful comments, and for your support of our paper. We appreciate that you found our problem setting well-motivated, our proposed framework clean, practically strong, and adaptable, and our paper easy to read. We first address your key questions below, and then your comments.
>
> **Q1**. Thank you for this question.
> * (i) For both domains, the initial context designs were sampled from the dataset of evaluations for the test task, uniformly from the subset of designs that achieved $\leq 30 \\%$ of the maximum reward.
> * (ii) For each test task, each repeated run for the task uses a different random seed for selecting the initial context designs. The random seeds are calculated based on the test task index and the run index.
> * (iii) The seeding is fixed across all methods, so all methods are evaluated with the exact same initial contexts. The candidate pool is always the same.
>
> **Q2.** For LCBench, we use a purely exploitative acquisition function, i.e. $\alpha(x) = \hat{\mu}(x)$. This is because the size of LCBench is most conducive to learning accurate point prediction over correct calibration.
>
> **Q3.** Thank you for this point. Our results are statistically significant, and we will include error bars on all plots. Below are the results for average normalized reward for both domains (Fig. 5a and Fig. 7a). We show the difference between our method and all competitive baselines across optimization, with $95 \\\%$ confidence intervals. For both domains, none of the confidence intervals overlap with 0, indicating that our method’s improvement is statistically significant.
>
> Gripper domain:
>
> | Trial | 1 | 5 | 10 | 15 | 20 | 25 | 30 |
> | :--- | :---: | ---: | :--- | :---: | ---: | :--- | :--- |
> | Ours - Ours w/o h | 0.048 $\pm$ 0.020 |  0.034 $\pm$ 0.014 | 0.032 $\pm$ 0.013 | 0.038 $\pm$ 0.011 | 0.037 $\pm$ 0.011 | 0.042 $\pm$ 0.011 | 0.042 $\pm$ 0.011 |
>
> Hyperparameter domain (scaled by 100 to %-range):
>
> | Trial | 1 | 5 | 10 | 15 | 20 | 25 | 30 |
> | :--- | :---: | ---: | :--- | :---: | ---: | :--- | :--- |
> Ours  - Ours w/o h | 1.165 $\pm$ 0.538 | 0.351 $\pm$ 0.076 | 0.105 $\pm$ 0.079 | 0.190 $\pm$ 0.077 | 0.245 $\pm$ 0.079 | 0.265 $\pm$ 0.073 | 0.268 $\pm$ 0.072 |
> Ours - DGP | 1.519 $\pm$ 0.564 | 0.196 $\pm$ 0.099 | 0.229 $\pm$ 0.098 | 0.253 $\pm$ 0.089 | 0.324 $\pm$ 0.087 | 0.295 $\pm$ 0.083 | 0.247 $\pm$ 0.078 |
>
> Although we report the standard $95 \\%$ CIs here, the results above are significant at the **p<0.01** confidence level.
>
> **Q4.** In MF/MIS BO, a fidelity/source is a cheaper approximation of $f(x)$ itself, and a single fidelity is queried at each step. In contrast, $h(x)$ is a potentially high-dimensional output that has a very distinct structure from $f(x)$, and is obtained during each evaluation of $f(x)$. Methods for MF/MIS BO often assume a hand-crafted kernel that relates different fidelities/sources, while the proposed approach learns a representation $E_\theta(h(x))$ that captures useful information relevant to $f(x)$, presenting a more challenging learning problem.
>
> **Comments:**
>
> **Noisy/Uninformative h(x).** Please see our response to Reviewer Yp5e regarding this question.
>
> **Overhead/sample efficiency of h(x).** Thank you for this interesting point. We agree that in many robotics domains, there can be a cost to collecting $h(x)$. For robotics, most of this cost may be upfront in the engineering effort to set up the sensors and data collection pipelines. Subsequently, the additional cost of collecting and storing $h(x)$ during a trial may add only a slight overhead to the run-time of the trial. Preprocessing and encoding $h(x)$ was also fast for our domains on GPU hardware. We agree that practical overhead and reliability is an important point of discussion, and we will add it to the main paper.
>
> **Evaluation Setting.** This is an important question, as the aim is to generalize to a high dimensional design space. The main practical difficulty with a fully continuous search space is that it requires running a simulation (gripper domain) or neural network training run (HP tuning) at each optimization step, which can take several hours for a single BO campaign. For the gripper design domain, we conducted initial experiments with a small scale dataset. Subsequently, we moved towards a continuous design space by scaling up the data collection substantially, and creating a large dataset of evaluations for each design task. While optimization proceeds over this discrete set of designs, the optimization is challenging as these sets are very large. Concretely, the average number of designs in the gripper design task is ~4,300, and for the hyperparameter tuning task it is 2,000. Thus, these sets represent a fairly comprehensive exploration of the design space.

---

> > ### Author Rebuttal · Reviewer_iLyv · 2026-04-01
> >
> > Thank you to the authors for addressing all of my concerns in the discussion. Please include these clarifications in the revision, especially the confidence intervals for both the proposed method and the baselines for comparison figures, as this would significantly strengthen the paper.

---

> > > ### Author Response · Authors · 2026-04-07
> > >
> > > Thank you for your thoughtful review and for confirming that our response addressed your concerns. We will include the confidence intervals for the comparison figures in the revision, as well as the other clarifications mentioned in the discussion.

---

### Official Review · Reviewer_pnmb · 2026-03-10

**Soundness:** 4
**Presentation:** 3
**Significance:** 3
**Originality:** 2
**Overall Recommendation:** 4
**Confidence:** 4

**Summary:**

The authors address the phenomenon in real-world expensive black-box optimization problems where experiments produce not only f(x) but also high-dimensional auxiliary information h(x). They propose a method that predicts f(x) for unseen points using a few-shot context containing h(x), aiming to improve Bayesian optimization performance. The authors also introduce, for the first time, a robotic gripper design task based on tactile feedback and evaluate their method on this task as well as on neural network hyperparameter tuning. Experimental results show that, with only a few samples, the method significantly outperforms several multi-task baselines in predicting the rewards f(x) of unseen designs.

**Compliance With Llm Reviewing Policy:**

Affirmed.

**Key Questions For Authors:**

1. In Section 6, when introducing the baselines, the paper states that the “Ours w/o h” variant uses an MLP to encode the context without h(x), whereas the original model employs a Transformer encoder. If the goal is to verify the effectiveness of h(x), shouldn’t the “Ours w/o h” baseline also use a Transformer encoder?
2. During the testing phase, the model parameters remain fixed. As Bayesian optimization proceeds, new points x and their corresponding observations f(x) are obtained, and these new x and f(x) are added to the context dataset. However, the paper does not provide experimental or theoretical analysis on whether adding new samples in this process may implicitly improve prediction accuracy.

**Limitations:**

The proposed method cannot handle new tasks from a distribution different from the training tasks, and the paper does not evaluate its generalization ability.

**Strengths And Weaknesses:**

Strengths：
1. The authors introduce, for the first time, a robotic gripper design task based on tactile feedback, and systematically leverage the high-dimensional auxiliary information h(x) generated during experiments to improve few-shot prediction and optimization performance, which has strong practical significance for real-world expensive black-box optimization.
2. The method is thoroughly evaluated and demonstrates significant effectiveness across both tasks.

Weaknesses：
See questions and limitations

---

> ### Author Rebuttal · Authors · 2026-03-29
>
> Thank you for your thoughtful comments and for your support of our paper. We appreciate that you found our proposed approach of strong practical significance, our evaluation thorough, and our results demonstrating effectiveness. We address your questions below.
>
> **"Ours w/o h” baseline.** In the original model (“Ours”), the context encoder encodes $h(x)$ using a sequence transformer, since $h(x)$ is a sequence over time. Separately, an MLP is used to encode $(x, f(x))$, which is non-sequential, and this is added to the encoding of $h(x)$ to form the final context encoding (shown in Fig. 2b). Finally, the encoded context point is passed into the few-shot prediction transformer architecture (Fig. 2a). The “Ours w/o h” baseline only removes the sequence transformer that encodes $h(x)$, since $h(x)$ is not provided in the context. Notably, all other parts of the model stay exactly the same, including the MLP encoder for $(x, f(x))$ and the transformer architecture for few-shot reward prediction. Since $(x, f(x))$ is non-sequential information, an MLP is more appropriate for encoding this information, so we use an MLP encoder for $(x, f(x))$ in both the “Ours w/o h” baseline and the original model.
>
> **Updating the context at test time.** We were a bit confused by this comment. Our paper answers your question, since our prediction results in Fig. 4 and Fig. 6 show that as the context size increases, the prediction accuracy for unseen points correspondingly increases. It is unclear to us what additional evidence is being requested in this regard. If the question refers to updating model parameters at test time, we agree with you that this is a possibility. However, the number of data-points at test time is very small (10-20 data-points), compared to the size of the training set (~1M data-points). Therefore, updating model parameters with test-time data may not have a positive effect on the model’s overall accuracy.

---

> > ### Author Rebuttal · Reviewer_pnmb · 2026-04-03
> >
> > My questions have been addressed.

---

> > > ### Author Response · Authors · 2026-04-07
> > >
> > > Thank you for your thoughtful review and for confirming that our response addressed your concerns.

---

### Official Review · Reviewer_Yp5e · 2026-03-10

**Soundness:** 2
**Presentation:** 2
**Significance:** 3
**Originality:** 3
**Overall Recommendation:** 4
**Confidence:** 3

**Summary:**

This paper studies a more realistic few-shot design optimization setting where each expensive evaluation returns not only a scalar reward $ f(x) $, but also auxiliary information $ h(x) $, such as tactile feedback or learning curves. To leverage such information, the authors propose a transformer-based few-shot surrogate that conditions on a small set of observed $ (x, f(x), h(x)) $ tuples and predicts the performance of unseen designs, which is then used within a Bayesian optimization loop. Experiments on robotic gripper design and hyperparameter optimization show that incorporating auxiliary information can improve both few-shot prediction and optimization efficiency.

**Compliance With Llm Reviewing Policy:**

Affirmed.

**Final Justification:**

The authors’ rebuttal addressed my concerns.

**Key Questions For Authors:**

- Q1. **How should the paper’s main contribution be understood: as a new Bayesian optimization method, or primarily as a stronger surrogate prediction model?**
   The current presentation positions the work as a BO contribution, but much of the method and evaluation seem to center on improving reward prediction by incorporating auxiliary information. The paper would be much clearer if the authors explicitly clarified what they consider to be the main methodological novelty.

- Q2. **How robust is the proposed approach when the auxiliary information is noisy, weakly informative, or only partially shared across tasks?**
   The current experiments are conducted in settings where the auxiliary signals appear highly structured and naturally informative. Could the authors provide stronger evidence, or at least a more substantive discussion, on whether the method would still work when such signals are less reliable or less transferable?

- Q3. **Why does the empirical BO evaluation not more directly test uncertainty quality and exploration capability, which are central to Bayesian optimization?**
   In its current form, the empirical study seems to emphasize predictive accuracy more than the core BO questions of uncertainty estimation and exploration–exploitation tradeoff. This makes it difficult to judge whether the gains truly reflect a better BO strategy rather than simply a better point predictor.

**Limitations:**

yes

**Strengths And Weaknesses:**

## Strengths

 1. **The paper studies a practically meaningful problem setting.** It extends few-shot design optimization to scenarios where each evaluation provides not only a scalar objective value but also auxiliary observations, which is well motivated in real applications.

 2. **The proposed framework is technically reasonable.** Using a transformer-based surrogate to encode auxiliary signals is a natural and coherent design choice for handling high-dimensional observations such as tactile sequences or learning curves.

 3. **The empirical results are generally encouraging.** Experiments on robotic gripper design and hyperparameter optimization suggest that auxiliary information can improve both prediction quality and optimization performance.

 4. **The benchmark construction adds value.** The newly introduced gripper benchmark appears to be a useful experimental asset for future research in this direction.

## Weaknesses

1. **The technical novelty appears limited relative to the claimed contribution.**
      At the method level, the paper mainly combines a transformer-based few-shot predictor with auxiliary-information encoding, and then plugs the learned predictor into a largely standard BayesOpt loop. The optimization procedure itself remains conventional: the next point is selected by maximizing an acquisition function over the surrogate prediction, and the main novelty is concentrated in the surrogate architecture rather than in a genuinely new BO principle.

2. **The paper does not fully establish that the proposed optimization framework, rather than simply a stronger predictor, is what drives the gains.**
   The training objective is explicitly to improve prediction of $ f(x) $ from context containing $ h(x) $, and the evaluation also emphasizes prediction accuracy as a primary metric. This makes the work feel closer to a representation-learning-for-surrogates paper than a fundamentally new optimization paper.

3.  **The BO evaluation is not as strong as it could be.**
   In the LCBench optimization experiments, the paper explicitly curates a subset of tasks with unique optima and uses a purely exploitative acquisition function, arguing that the benchmark is better suited to accurate point prediction than calibration. While understandable, this choice also makes the BO evaluation less comprehensive, because it avoids testing whether the method provides robust uncertainty estimates or improves the exploration–exploitation tradeoff in more standard BO settings.

---

> ### Author Rebuttal · Authors · 2026-03-29
>
> Thank you for your detailed and thoughtful comments, and for your support of our paper. We appreciate that you found our problem setting practically motivated, our method natural and reasonable, our empirical results encouraging, and our benchmark useful for future work. We address your questions below.
>
> **Main Contribution and Novelty.** We agree that our primary contribution is a stronger surrogate model. This surrogate model is motivated by BayesOpt, and we show that it provides strong empirical results when applied to multiple optimization problems. Thank you for raising this helpful point, and we will modify the paper to clarify the contribution. We note that the design of the surrogate model is informed by the optimization setting. While there are many methodological choices for how to use $h(x)$, using it as conditioning information for predicting reward encourages the model to preserve aspects of $h(x)$ most relevant for optimization.
>
> **Noisy/weakly informative auxiliary information.** Thank you for raising this interesting point. Our approach is unlikely to overfit to noise or spurious/uninformative features in $h(x)$, since such features are unlikely to be useful across all training tasks. Thus, the multi-task training provides robustness and encourages the model to extract whatever predictive signal is consistently present across all data-points and all training tasks. In the limiting case where $h(x)$ is completely uninformative, the model would learn to disregard $h(x)$ and only rely on $(x, f(x))$ for prediction, achieving the accuracy of the "Ours w/o h" baseline as a lower bound.
>
> We note that for the gripper design task, the tactile information is quite sparse. For a 16x16 tactile image, usually no more than 3-4 pixels have non-zero contact readings. Even though the useful signal is quite sparse, our method still leverages $h(x)$ successfully to accelerate optimization of test tasks. We did not study the effect of adding progressive noise to $h(x)$, and we agree that this is an interesting study which we could investigate in future work.
>
> **Empirical BO Evaluation of Uncertainty.** We agree that our results do not extensively test uncertainty quantification. For the gripper design domain, our surrogate model with Probability of Improvement outperforms ours with a purely exploitative acquisition function. This suggests that given enough training tasks, our model can learn useful uncertainty estimates that support effective optimization of unseen tasks.

---

> > ### Author Rebuttal · Reviewer_Yp5e · 2026-04-01
> >
> > Thank you to the authors for addressing all of my concerns in the discussion.

---

> > > ### Author Response · Authors · 2026-04-07
> > >
> > > Thank you for your thoughtful review and for confirming that our response addressed your concerns.

---

### Decision · Program_Chairs · 2026-04-30

**Decision:**

Accept (regular)

**Comment:**

This paper introduces a novel and practically relevant setting for few-shot design optimization, addressing scenarios where experimental evaluations yield not only a scalar reward $f(x)$ but also high-dimensional auxiliary information $h(x)$. To tackle this, the authors propose a transformer-based surrogate model that conditions on this auxiliary feedback from a history of related tasks to accelerate optimization.

The paper has several key strengths:
* **Practical Relevance and Novelty:** All reviewers pointed out the high relevance of problem framing, noting its direct applicability to real-world expensive black-box problems like robotics and hyperparameter tuning.
* **Strong Empirical Results:** The method demonstrates clear empirical gains, consistently outperforming several relevant multi-task baselines.
* **New Benchmarks:** The paper introduces a large-scale robotic gripper benchmark dataset, which is a highly valuable asset for future research in the community.

Given the meaningful setting, the solid technical execution, the clear empirical benefits, the significant benchmark contribution, and the successful resolution of reviewer concerns during the rebuttal, the consensus is to accept the paper.

**Note to Authors:** Please incorporate the promised revisions into the camera-ready version. In particular, please include confidence intervals on all plots and add the expanded discussion analyzing the runtime overhead and practical costs of collecting and processing the auxiliary information, as requested during the discussion phase.